# The β-TrCP-FBXW2-SKP2 axis regulates lung cancer cell growth with FBXW2 acting as a tumour suppressor

Jie Xu[1], Weihua Zhou[1], Fei Yang[2], Guoan Chen[3], Haomin Li[2,4], Yongchao Zhao[1,2,5], Pengyuan Liu[2,6], Hua Li[1], Mingjia Tan[1], Xiufang Xiong[2] & Yi Sun[1,2,7]

β-TrCP and SKP2 are two well-studied F-box proteins, which often act as oncogenes. Whether and how they communicate with each other is unknown. Here we report that FBXW2, a poorly characterized F-box, is a substrate of β-TrCP1 and an E3 ligase for SKP2. While β-TrCP1 promotes FBXW2 ubiquitylation and shortens its half-life, FBXW2 does the same to SKP2. FBXW2 has tumour suppressor activity against lung cancer cells and blocks oncogenic function of both β-TrCP1 and SKP2. The levels of β-TrCP1-FBXW2-SKP2 are inversely correlated during cell cycle with FBXW2 and β-TrCP/SKP2 being high or low, respectively, in arrested cells, whereas the opposite is true in proliferating cells. Consistently, FBXW2 predicts a better patient survival, whereas β-TrCP1 and SKP2 predict a worse survival. Finally, the gain- and loss-of-function mutations of FBXW2 are found in various human cancers. Collectively, our data show that the β-TrCP-FBXW2-SKP2 axis forms an oncogene-tumour suppressor-oncogene cascade to control cancer cell growth with FBXW2 acting as a tumour suppressor by promoting SKP2 degradation.

[1] Division of Radiation and Cancer Biology, Department of Radiation Oncology, University of Michigan, Ann Arbor, Michigan 48109, USA. [2] Institute of Translational Medicine, Zhejiang University School of Medicine, Hangzhou 310029, China. [3] Department of Surgery, University of Michigan, Ann Arbor, Michigan 48109, USA. [4] Affiliated Children Hospital, Zhejiang University School of Medicine, Hangzhou 310003, China. [5] Key laboratory of combined multi-organ transplantation, Ministry of Public Health, First Affiliated Hospital, Zhejiang University School of Medicine, Hangzhou 310058, China. [6] Sir Run Run Shaw Hospital, Zhejiang University School of Medicine, Hangzhou 310020, China. [7] Collaborative Innovation Center for Diagnosis and Treatment of Infectious Diseases, Zhejiang University, Hangzhou 310058, China. Correspondence and requests for materials should be addressed to Y.S. (email: sunyi@umich.edu or yisun@zju.edu.cn).

Lung cancer, including small cell lung cancer and non-small cell lung cancer (NSCLC), is the leading cause of cancer-related death in the USA and around the world[1]. NSCLC, as the most common type of lung cancer, represents more than 80% of the total cases[2]. Among the molecular changes found in NSCLC, mutational activation of Kras and EGFR, and mutational inactivation of p53 are the most common genetic alterations[3,4]. Recently, dysregulation of various components of the ubiquitin-proteasome system, which maintains the protein homeostasis by timely degradation of unwanted proteins[5], has been implicated in lung cancer[6].

The SCF (SKP1-Cullins-F box proteins) E3 ubiquitin ligase, also known as CRL1 (Cullin-RING ligase 1), is the largest family of E3 ubiquitin ligase. Each individual SCF E3 ligase consists of adaptor protein SKP1, cullin-1, one F-box protein out of ~69 family members, and one out of two RING family proteins RBX1/ROC1 or RBX2/ROC2/SAG/RNF7 (refs 7–10). Among SCF components, cullin 1 acts as a molecular scaffold that simultaneously interacts with the SKP1-F-box complex at its N terminus and RBX1 or RBX2 at its C terminus[11]. The F-box proteins bind to SKP1 and cullin through the F-box domain, and to substrates through their WD40 or leucine-rich repeats. While F-box proteins determine the substrate specificity of the SCF complex[11,12], the cullin-RBX1/2 complex, with RBX1/2 binding to ubiquitin loaded E2, constitutes the core E3 ligase activity[13]. The SCF E3 ubiquitin ligases, by promoting timely ubiquitylation and subsequent degradation of more than 350 substrates, control several important biological processes[14–17]. Abnormal regulation of SCF E3 results in uncontrolled proliferation, genomic instability and cancer[7,15,18–20].

F-box proteins emerge as important players in tumorigenesis[21]. β-TrCP1 (β-transducin repeat-containing protein 1), one of the prototypical and best characterized mammalian F-box proteins, exerts oncogenic functions in most cases by promoting targeted degradation of many tumour suppressors, including p53 (ref. 22), Bim EL (ref. 23), procaspase-3 (ref. 24), IκB (ref. 25), programmed cell death protein 4 (PDCD4) (ref. 26) and DEPTOR (refs 27–29). Consistently, β-TrCP1 was overexpressed and associated with poor prognosis in colorectal cancer[30], pancreatic cancer[31], hepatoblastomas[32] and breast cancer[33]. Like β-TrCP1, SKP2, another well-characterized F-box protein, acts as a classic oncogene which promotes proliferation and survival of cancer cells, mainly through targeted degradation of a number of tumour suppressive proteins, including p21 (ref. 34), p27 (refs 35,36), p57 (ref. 37), p130 (ref. 38), FOXO1 (ref. 39), among many others[40]. In NSCLC, Skp2 overexpression increased the capacity of invasion[41], and is associated with aggressiveness[42], metastasis[43] and poor prognosis[44]. FBXW7, on the other hand, is a well-established tumour suppressor that targets various oncogenic proteins for degradation[45]. Except for these three well-known F-box proteins, potential roles of remaining 66 F-box proteins in cancers, particularly in lung cancer, are poorly understood, although some of them are involved in normal physiology and disorders in the lung such as inflammatory lung disease[46]. Moreover, it is largely unknown whether and how F-box proteins regulate each other via targeted degradation to control proliferation and survival of lung cancer cells.

Herein, we demonstrated that FBXW2, a poorly characterized F-box protein, is a novel substrate of β-TrCP1, and a novel E3 ligase of SKP2 for targeted degradation. The levels of β-TrCP1-FBXW2-SKP2 are inversely regulated in a coordinated manner during cell cycle progression. We found that in contrast to oncogenic β-TrCP1 and SKP2, FBXW2 acts as a tumour suppressor to inhibit growth and survival of lung cancer cells, and high FBXW2 expression predicts a better patient survival.

We also found FBXW2 point mutations in human cancer with gain- or loss-of-function activity. Our study established a previously unknown signalling cascade of the β-TrCP-FBXW2-SKP2 by forming the oncogene (β-TrCP)-tumour suppressor gene (FBXW2)-oncogene (SKP2) axis that regulates growth and survival of lung cancer cells via targeting each other for degradation.

## Results

### β-TrCP1 binds to FBXW2 and negatively regulates FBXW2 levels.

Although β-TrCP1 has been shown to bind to and ubiquitylate many cellular proteins for targeted degradation[47,48], very few are F-box proteins[49]. Recently, a large-scale proteomic study identified FBXW2 as a potential binding partner of β-TrCP1 (ref. 50). To determine potential binding of other F-box proteins with β-TrCP1, we transfected 7 F-box proteins (FBWX2, FBXW4, FBXW5, FBXW7, FBXW8, FBXL3 and SKP2) into 293 cells, followed by immunoprecipitation assay to detect endogenous β-TrCP1. We found that ectopically expressed FBXW2 pulled-down the high level of endogenous β-TrCP1, even expressed at the lowest level (Supplementary Fig. 1a), suggesting that FBXW2 may be subjected to β-TrCP1-mediated degradation. To further determine whether FBXW2 is a substrate of SCF E3, we treated lung cancer A549 cells with MLN4924, a small molecule inhibitor of NEDD8-activating enzyme, which inactivates SCF E3 by inhibiting cullin-1 neddylation[51], and found MLN4924 causes a dose-dependent accumulation of FBXW2 (Supplementary Fig. 1b). Moreover, we searched the FBXW2 sequence for the β-TrCP consensus binding motif (DSGXXS) and found an evolutionarily conserved putative binding site (SSGART) on codons 214–219 (Supplementary Fig. 1c). Subsequent immunoprecipitation assays showed that βTrCP-FBXW2 binding can be detected under ectopic overexpressed conditions (Supplementary Fig. 1d), as well as under unstimulated physiological conditions, with no binding found between βTrCP and FBXL3 or FBXL11, which were served as specificity controls (Fig. 1a and Supplementary Fig. 1e). Finally, we found that β-TrCP-FBXW2 binding is phosphor-dependent, since β-TrCP only binds to a FBXW2 peptide with β-TrCP binding motif phosphorylated (pSpSGARpT) (Fig. 1b). These results established an *in vivo* interaction between β-TrCP1 and FBXW2.

Having detected a physical interaction between two proteins, we next determined whether FBXW2 protein level is regulated by β-TrCP1. In a co-transfection experiment, the levels of ectopically expressed FBXW2 were reduced in a dose-dependent manner upon β-TrCP1 co-transfection (Fig. 1c and Supplementary Fig. 1f). A similar dose-dependent reduction of endogenous FBXW2 was also detected when β-TrCP1 was transfected alone (Fig. 1d and Supplementary Fig. 1g). While overexpression of wild-type β-TrCP1 reduced FBXW2 levels in a dose-dependent manner, overexpression of β-TrCP1ΔF, a dominant-negative mutant that binds to the substrates but unable to recruit other components of SCF E3 ligase[24,27], had no effect (Fig. 1e and Supplementary Fig. 1h). Furthermore, overexpression of β-TrCP had no effect on the level of FBXW2 mRNA (Fig. 1f), suggesting β-TrCP promotes FBXW2 degradation. Finally, we performed an siRNA-based knockdown experiment in A549 and H23 lung cancer cells, and found endogenous levels of FBXW2 protein, but not FBXW2 mRNA were increased upon β-TrCP1 knockdown (Fig. 1g). We then determined whether β-TrCP1 shortened FBXW2 protein half-life. Indeed, while transfected FBXW2 remained stable after 8 h of cycloheximide treatment, β-TrCP1 co-transfection significantly reduced FBXW2 basal level and shortened its half-life (Fig. 1h and Supplementary Fig. 1i). No effect was found when β-TrCP1ΔF mutant was

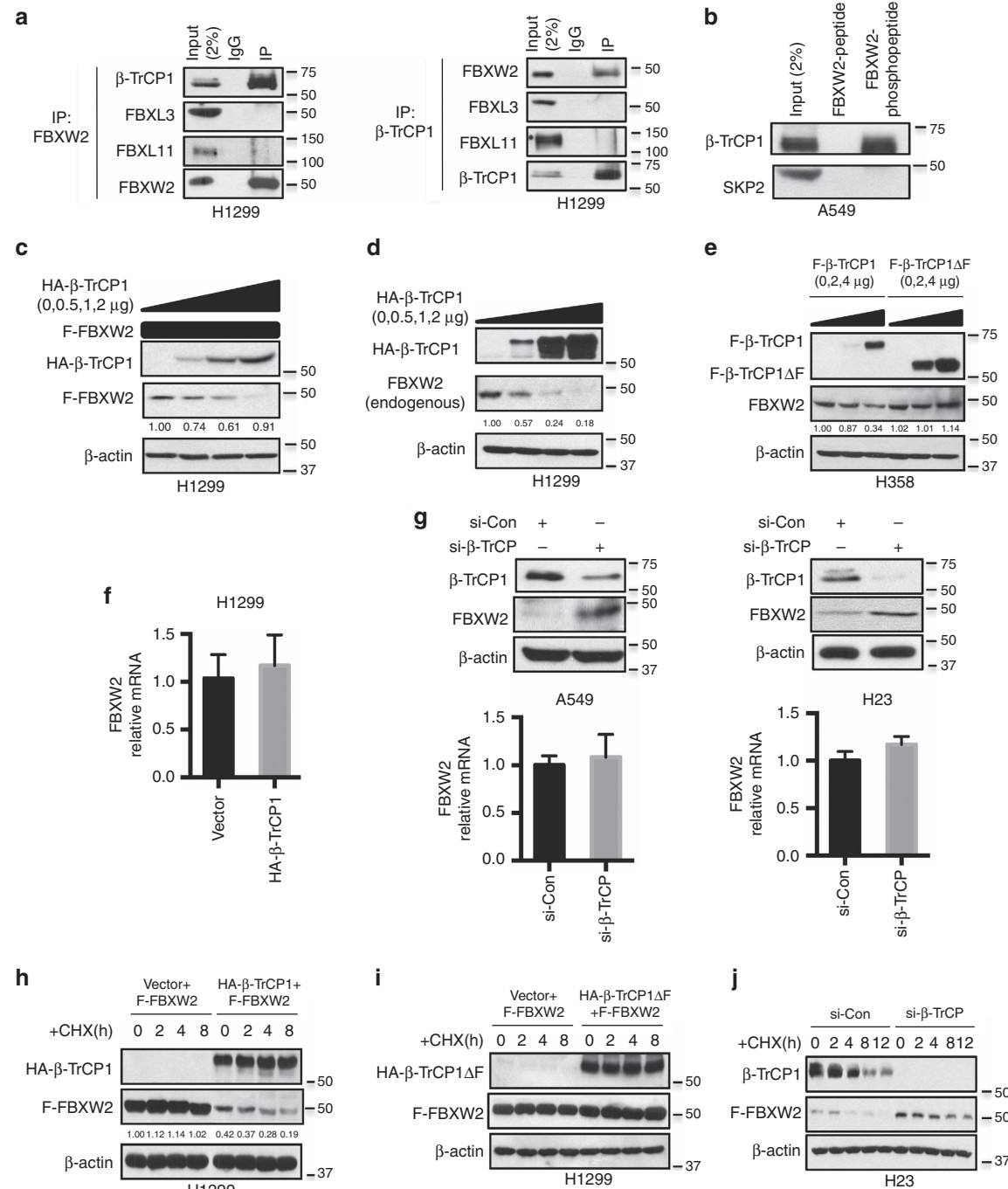

**Figure 1 | β-TrCP1 binds to FBXW2 and negatively regulates FBXW2 levels.** (**a**) β-TrCP1 binds to endogenous FBXW2: Cell lysates from H1299 cells were pulled down with anti-FBXW2 or anti-β-TrCP1 Abs, followed by IB with indicated Abs. (**b**) β-TrCP1 binds to phosphor-FBXW2 peptide: Cell lysates from A549 cells were incubated with bead-conjugated FBXW2 non-phosphorylated or phosphor-peptide containing β-TrCP binding motif. Beads were washed and subjected to IB with indicated Abs. (**c–e**) Overexpression of β-TrCP1, but not its ΔF mutant, decreases the levels of FBXW2 protein: H1299 and H358 cells were co-transfected with F-FBXW2 (FLAG-tagged FBXW2) and increasing amounts of β-TrCP1, or transfected with increasing amounts of β-TrCP1 or β-TrCP1ΔF alone, followed by IB with indicated Abs. The band density was quantified. (**f**) β-TrCP1 overexpression has no effect on FBXW2 mRNA: H1299 cells were transfected with β-TrCP1 or vector control, followed by qRT-PCR for mRNA expression. Error bars indicate mean + s.d. of three repeats. (**g**) β-TrCP silencing increases the endogenous levels of FBXW2 protein, but not mRNA: A549 and H23 cells were transfected with siRNA targeting both β-TrCP1 and β-TrCP2, along with scrambled siRNA, followed by IB (top panels) or qRT-PCR (bottom panel). Error bars indicate mean + s.d. of three repeats. (**h,i**) FBXW2 half-life is shortened by β-TrCP1, but not by β-TrCP1ΔF: FLAG-FBXW2 was transfected into H1299 cells, along with the vector control or plasmid expressing HA-β-TrCP1 or HA-β-TrCP1ΔF. Cells were switched to fresh medium (10% FBS) containing cycloheximide (CHX) 48 h post transfection for indicated time periods and harvested for IB. The band density was quantified. (**j**) β-TrCP1 RNAi silencing extends protein half-life of endogenous FBXW2. H23 cells were transfected with either control RNAi, or RNAi targeting both β-TrCP1 and β-TrCP2 for 48 h. Cells were cultured in fresh medium containing CHX and incubated for indicated time periods before being harvested for IB.

transfected (Fig. 1i and Supplementary Fig. 1j). On the other hand, transfection of siRNA targeting both β-TrCP1 and β-TrCP2 extended FBXW2 protein half-life, leading to its stabilization (Fig. 1j). Collectively, these data strongly suggest that FBXW2 could be a novel substrate of β-TrCP1.

**β-TrCP1 promotes FBXW2 ubiquitylation and degradation.** We further generated an FBXW2 mutant on β-TrCP binding motif with serine and threonine residues all mutated to alanine ('SSGART' to 'AAGARA', designated as FBXW2-3A) and found that β-TrCP1 no longer had any effect on the basal

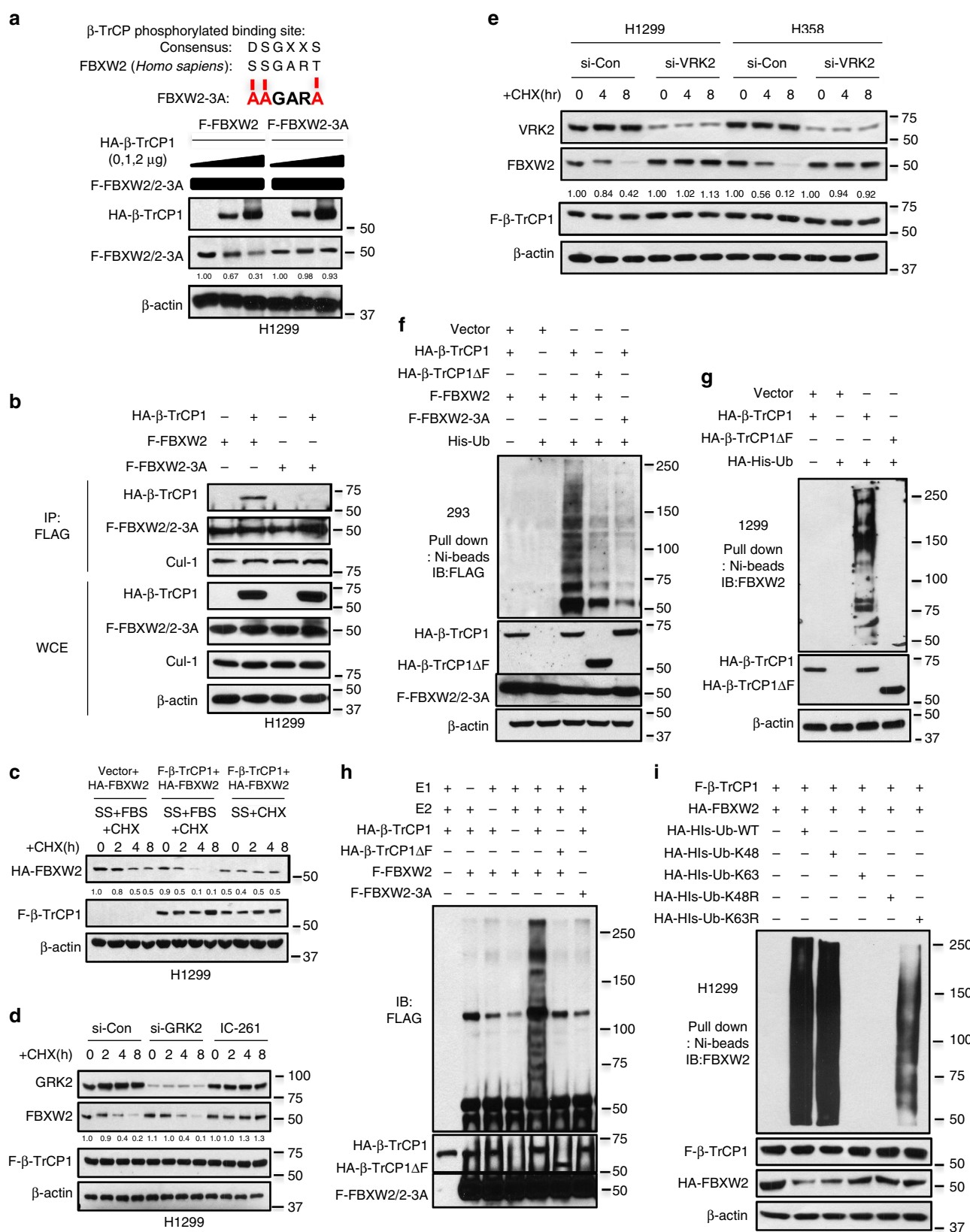

level of FBXW2-3A (Fig. 2a), nor on its half-life (Supplementary Fig. 2a), indicating that targeted degradation of FBXW2 by β-TrCP1 is totally dependent on β-TrCP1 binding motif. Furthermore, we found that unlike wild-type FBXW2, ectopically expressed FBXW2-3A failed to pull-down exogenous β-TrCP1, although it was able to pull-down cullin-1 (Fig. 2b). Likewise, ectopically expressed β-TrCP1 failed to pull-down FBXW2 mutant, FBXW2-3A (Supplementary Fig. 2b), indicating the degradation of FBXW2 by β-TrCP1 is binding dependent.

It is well-established that phosphorylation of a substrate at its F-box binding motif is prerequisite in most cases for its binding to F-box protein for subsequent ubiquitylation and degradation[14]. We previously found that serum addition to serum-starved cells could activate kinase and trigger substrate phosphorylation[27]. We, therefore, added serum to serum-starved cells and found that ectopically expressed β-TrCP1 significantly shortened the protein half-life of exogenous FBXW2 in H1299 (Fig. 2c) or H358 lung cancer cells (Supplementary Fig. 2c). To identify potential kinase(s) that would mediate FBXW2 phosphorylation at the β-TrCP1 binding motif (SSGART), we searched computer database (GSP 3.0 http://gps.biocuckoo.org) for consensus kinase binding site, and identified CK1 (casein kinase), VRK2 (vaccinia-related kinase-2), a serine/threonine kinase similar in sequence and structure to the catalytic domain of the CK1 family[52] and GRK2 (G protein-coupled receptor kinase) as candidates with the highest score (Supplementary Table 1). We followed this lead, and inactivated VRK2 and GRK2 by siRNA silencing, or CK1 and VRK2 by small molecule inhibitor IC-261 (refs 53,54), and found that GRK2 knockdown had no effect, whereas CK1/VRK2 inhibition or VRK2 silencing largely blocked the degradation of exogenously expressed FBXW2 (Fig. 2d,e and Supplementary Fig. 2d). Consistently, IC-261 treatment, or VRK2 silencing, but not GRK2 silencing, abrogated β-TrCP1 and FBXW2 binding (Supplementary Fig. 2e,f). Collectively, CK1 and VRK2, but not GRK2 kinase, appears to mediate FBXW2 phosphorylation at the β-TrCP binding motif.

We next determined whether β-TrCP1 would promote FBXW2 poly-ubiquitylation using classic in vivo and in vitro ubiquitylation assays. The in vivo ubiquitylation assay showed that wild-type β-TrCP1, but not its β-TrCP1ΔF mutant, significantly promoted poly-ubiquitylation of exogenously expressed as well as endogenous FBXW2 (Fig. 2f,g and Supplementary Fig. 2g). Consistently, β-TrCP1 only promoted poly-ubiquitylation of wild-type FBXW2, but not FBXW2-3A with β-TrCP1 binding site abrogated (Fig. 2f and Supplementary Fig. 2h). Similarly, the in vitro poly-ubiquitylation assay, using an in vitro purified system containing E1, E2, E3s

(β-TrCP1 or β-TrCP1ΔF) and substrates (FBXW2 or FBXW2-3A) showed that β-TrCP1, but not its ΔF mutant, significantly promoted poly-ubiquitylation of wild-type FBXW2, but had no effects on FBXW2-3A mutant (Fig. 2h). Finally, using various ubiquitin mutants, we confirmed that FBXW2 polyubiquitylation by β-TrCP1 is mediated via K-48 linkage (Fig. 2i). Taken together, our results support the notion that FBXW2 is a substrate of β-TrCP1, which binds to FBXW2 via its binding motif, and promotes its ubiquitylation for subsequent degradation.

**FBXW2 suppresses growth and survival of lung cancer cells.** We next determined the biological function of FBXW2 against lung cancer cells. Ectopic expression of FBXW2 in H23 and H358 lung cancer cells caused a significant reduction in monolayer growth, clonogenic survival and the anchorage-independent growth (Fig. 3a–c and Supplementary Fig. 3a–b). Likewise, FBXW2 knockdown in H1299 and H358 cells promoted monolayer growth, clonogenic survival and the anchorage-independent growth (Fig. 3d–f and Supplementary Fig. 3c–d). To elucidate the mechanism by which FBXW2 suppressed proliferation and survival of lung cancer cells, we measured, after manipulation of FBXW2, the levels of few tumour suppressive and oncogenic proteins, known to regulate cell growth as well as known to be substrates of F-box proteins. We found that over-expression of FBXW2, but not FBXW4 nor FBXW8 (serving as negative controls), induced accumulation of several tumour suppressive proteins, including p21, p27, p130 and FOXO1 (Fig. 3g and Supplementary Fig. 3e), in a dose-dependent manner (Fig. 3h). Consistently, FBXW2 knockdown caused reduction of these proteins (Fig. 3i and Supplementary Fig. 3f). Interestingly, neither FBXW2 overexpression nor depletion had any effect on the levels of oncogenic proteins, including WEE-1, c-JUN and Notch-1 (Fig. 3g–i and Supplementary Fig. 3e,f). Collectively, these results strongly suggest that FBXW2 has tumour suppressor activity against lung cancer, which is likely mediated by accumulation of few tumour suppressor proteins.

**Characterization of SKP2 as a new substrate of FBXW2.** Given that tumour suppressor proteins p21, p27, p130 and FOXO1 are all known substrates of SKP2 (ref. 47), we tested a hypothesis that SKP2 might be a novel substrate of FBXW2. We therefore first determined potential binding of multiple F-box proteins with endogenous SKP2 and found a strong binding between FBXW2 and SKP2 and a weak binding between β-TrCP1 and SKP2, which could be mediated by FBXW2 (Supplementary Fig. 4a). We next determined whether two proteins bind to each other under physiological conditions. Indeed, using

**Figure 2 | β-TrCP1 binding to its degron sequence on FBXW2 and promotes FBXW2 ubiquitylation.** (**a**) β-TrCP1 has no effects on FBXW2 binding mutant: H1299 cells were transfected with indicated plasmids, switched 48 h later to fresh medium containing CHX and harvested at indicated periods for IB. The band density was quantified. (**b**) Loss of β-TrCP1-FBXW2 binding in degron site mutant: FBXW2 or its 3A mutant at the degron site was co-transfected with HA-β-TrCP1, followed by IP with FLAG Ab and IB. (**c**) β-TrCP1 shortens the protein half-life of FBXW2, triggered by serum addition to serum-starved (SS) cells: H1299 cells were transfected with indicated plasmids, switched 48 h later to fresh medium containing 10% FBS (fetal bovine serum) and CHX and harvested at indicated periods for IB. The band density was quantified. (**d,e**) CK1 kinase mediates FBXW2 phosphorylation at β-TrCP1 binding motif: H1299 cells were transfected of siRNA targeting GRK2, or treated by CK1 inhibitor IC-261 (10 μM) (**d**), or transfected with siRNA targeting VRK2 (**e**), followed by transfected with FLAG-β-TrCP1 for 48 h. Cells were harvested at indicated points after CHX treatment for IB. The band density was quantified. (**f,g**) β-TrCP1 promotes FBXW2 ubiquitylation in vivo: 293 (**f**) or H1299 (**g**) cells were transfected with indicated plasmids, lysed under denatured condition at 6 M guanidinium solution, followed by Ni-beads pull-down. Washed beads were boiled for IB to detect polyubiquitylation of exogenous FBXW2 (**f**) or endogenous FBXW2 (**g**). (**h**) β-TrCP1 promotes FBXW2 ubiquitylation in vitro: β-TrCP1 (E3) was prepared by transfecting HA-β-TrCP1 or HA-β-TrCP1ΔF into 293 cells, followed by HA-bead IP and 3 × 3 HA peptide elution. FBXW2 and FBXW2 mutant were prepared by transfecting FLAG-FBXW2 or FLAG-FBXW2-3A into 293 cells, followed by FLAG-bead IP. β-TrCP1 (E3) and FBXW2 (substrate) were added into a reaction mixture containing ATP, ubiquitin, E1 and E2, followed by constant mixing for 60 min. The reaction mixture was then loaded onto PAGE gel for IB using anti-FLAG Ab. (**i**) β-TrCP1 promotes FBXW2 ubiquitylation via K48 linkage: H1299 cells were transfected with indicated plasmids, lysed under denatured condition at 6 M guanidinium solution, followed by Ni-beads pull-down and IB for FBXW2.

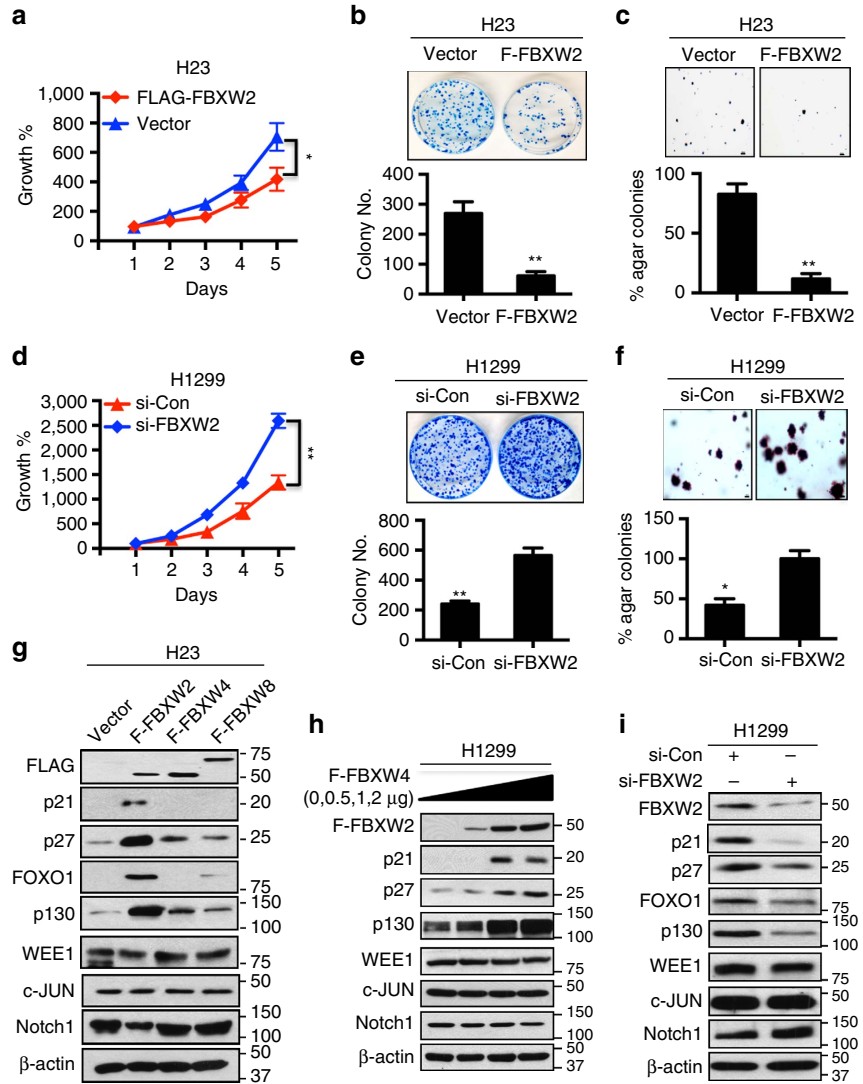

**Figure 3 | FBXW2 is a putative tumour suppressor against lung cancer cells.** (**a–f**) FBXW2 overexpression inhibits, but siRNA silencing promotes growth and survival of lung cancer cells: H23 cells were transfected with siRNA targeting FBXW2, along with the control siRNA (**a–c**), or H1299 cells were transfected with FLAG-FBXW2 and the vector control (**d–f**), followed by ATP-lite proliferation assay ($n=3$) (**a,d**); clonogenic survival ($n=3$) (**b,e**); and soft agar growth ($n=3$) (**c,f**). Scale bars, 100 μm (**c,f**). Shown is mean ± s.e.m. Student's $t$-test was performed, $*P<0.05$; $**P<0.01$. (**g–i**) FBXW2 overexpression increases, whereas siRNA silencing decreases the levels of tumour suppressor proteins: Cells were transfected with FLAG-FBXW2, FBXW2 or FBXW4 as negative controls, along with vector control (**g**), increasing amount of FLAG-FBXW2 (**h**), or siRNA oligonucleotide targeting FBXW2 (**i**), followed by IB with indicated Abs.

pull-down assay, we detected endogenous SKP2, but not FBXL3 or FBXL11 in FBXW2 immuno-precipitants and endogenous FBXW2, but not FBXL3 or FBXL11 in SKP2 immuno-precipitants, respectively (Fig. 4a). We next mapped binding domains between FBXW2 and SKP2, and found that the C-terminal WD40 domain of FBXW2 (codons 139–454) binds to N-terminal domain of SKP2 (codons 1–150) (Supplementary Fig. 4b,c). To further define FBXW2 binding motif on SKP2, we compared the first 150 amino acids of SKP2 to a phosphor-peptide (codons 313–333) of GCM1, known to bind to FBXW2 (ref. 55), with special attention to the motif containing a string of serine/threonine residues. Indeed, we found an evolutionarily conserved motif of TSXXXS on codons 29–34 of SKP2 (Supplementary Fig. 4d). To test whether this is the motif mediating FBXW2-SKP2 binding, we mutated all three TSS residues to alanine (TSS→AAA) and found that mutant SKP2-3A is no longer bound to FBXW2 (Fig. 4b). We then

determined the effect of FBXW2 on SKP2 levels and found that the levels of endogenous SKP2 protein, but not mRNA were reduced upon FBXW2 transfection in lung cancer cells (Fig. 4c,d). Furthermore, FBXW2 overexpression significantly reduced, in a dose dependent manner, the levels of exogenous (Fig. 4e and Supplementary Fig. 4e) as well as endogenous (Fig. 4f and Supplementary Fig. 4f) SKP2, which is largely abrogated by MG132 treatment (Fig. 4f), whereas FBXW2 depletion caused a significant accumulation of SKP2 protein without affecting SKP2 mRNA levels (Fig. 4g and Supplementary Fig. 4g), indicating that FBXW2 negatively regulates the SKP2 protein levels.

It was previously known that SKP2 is ubiquitylated and degraded by APC$^{CDH1}$ E3 ligase[56,57], we therefore determined whether FBXW2-induced SKP2 degradation is dependent or independent of CDH1. While CDH1 depletion caused a minimal, if any, effect on the levels of SKP2 in two lung cancer cell lines,

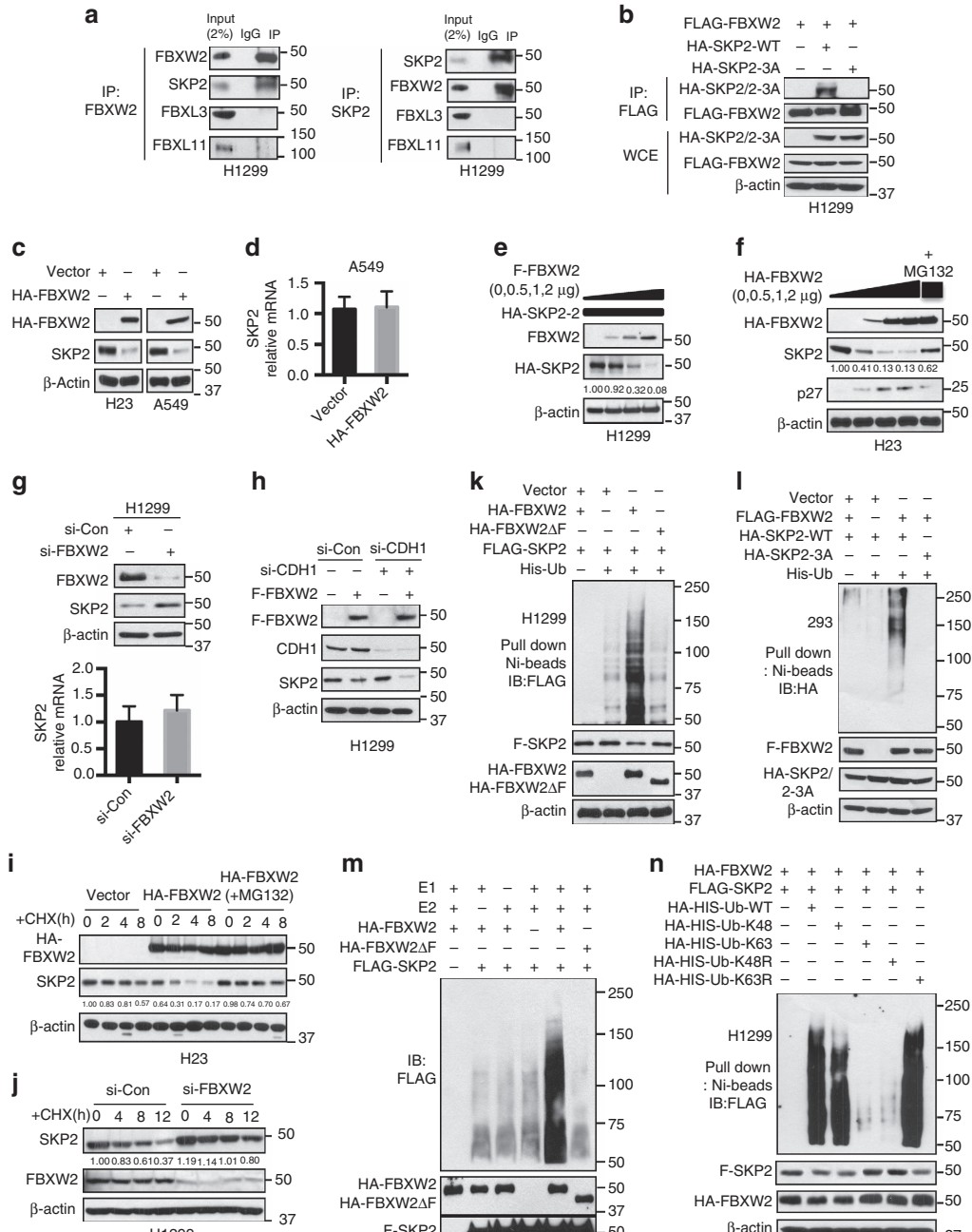

**Figure 4 | FBXW2 binds to SKP2 and promotes its ubiquitylation and degradation.** (**a**) FBXW2 binds to SKP2 *in vivo*: Lysates from H1299 cells were pulled down with anti-FBXW2 (left panel) or anti-SKP2 (right panel), followed by IB. (**b**) FBXW2 failed to bind to a SKP2 mutant: H1299 cells were tranfected with indicated plasmids, followed by FLAG IP and HA IB or direct IB. (**c,d**) Ectopically expressed FBXW2 reduces the endogenous levels of SKP2 protein, but not mRNA: Cells were transfected with HA-FBXW2, followed by IB (**c**) or qRT-PCR for SKP2 (**d**). Error bars indicate mean + s.d. of three repeats. (**e,f**) FBXW2 overexpression decreases the levels of the exogenous and endogenous SKP2 proteins: Cells were co-transfected with indicated plasmids, followed by IB 48 h post transfection. (**g**) FBXW2 depletion increases the levels of SKP2 protein, but not mRNA: Cells were transfected with FBXW2 siRNA and scramble siRNA, followed by IB or qRT-PCR for SKP2. Error bars indicate mean + s.d. of three repeats. (**h**) FBXW2-induced SKP2 degradation is independent of CDH1: Cells were transfected with FLAG-FBXW2 and/or siRNA against CDH1, and then harvested for IB. (**i**) FBXW2 shortens SKP2 half-life: Cells were transfected with HA-FBXW2, and switched 48 h post transfection to fresh medium containing CHX for indicated periods with or without MG132 treatment for last 2 h, and then harvested for IB. The band density was quantified. (**j**) FBXW2 knockdown extends SKP2 half-life: Cells were transfected with siRNA targeting FBXW2. Cells were switched 48 h later to fresh medium containing CHX for indicated periods and harvested for IB. The band density was quantified. (**k,l**) FBXW2 promotes ubiquitylation of SKP2 (**k**), but not SKP2-3A mutant (**l**): Cells were transfected with indicated plasmids, followed by Ni-beads pull-down and IB for SKP2. (**m**) FBXW2 promotes SKP2 ubiquitylation by *in vitro* assay: H1299 cells were transfected with indicated plasmids. Pull-down purified FBXW2 and FBXW2ΔF (E3s), Pull-down purified SKP2 (substrate), were added into a reaction mixture containing ATP, ubiquitin, E1 and E2, followed by IB using anti-FLAG Ab. (**n**) FBXW2 promotes SKP2 ubiquitylation via K48 linkage: Cells were transfected with indicated plasmids, lysed under denatured condition at 6 M guanidinium solution, followed by Ni-beads pull-down and IB for SKP2.

it did not affect FBXW2-induced SKP2 reduction at all (Fig. 4h and Supplementary Fig. 4h). Thus, degradation of SKP2 by FBXW2 is independent of CDH1.

We next determined whether FBXW2 regulated SKP2 protein half-life. While the vector control had no effect on endogenous SKP2, FBXW2 transfection decreased the basal level of SKP2 and shortened its protein half-life from >8 h to ∼4 h, which is completely abrogated by MG132 treatment (Fig. 4i and Supplementary Fig. 4i). Consistently, FBXW2 depletion by siRNA silencing significantly extended protein half-life of endogenous SKP2 (Fig. 4j and Supplementary Fig. 4j). Furthermore, the *in vivo* ubiquitylation assay showed that FBXW2, but not its ΔF mutant, promoted the poly-ubiquitylation of SKP2 (Fig. 4k and Supplementary Fig. 4k). In consistent with lack of binding between FBXW2 and SKP2-3A mutant, FBXW2 failed to promote its poly-ubiquitylation (Fig. 4l). Serving as negative controls, neither FBXW4 nor FBXW8 promoted polyubiquitylation of SKP2 (Supplementary Fig. 4l) or affected SKP2 levels (Supplementary Fig. 4m). Finally, the *in vitro* ubiquitylation assays also showed that FBXW2, but not its ΔF mutant, promoted the poly-ubiquitylation of SKP2 (Fig. 4m), and SKP2 polyubiquitylation by FBXW2 was mediated again via classic K48 linkage (Fig. 4n). Taken together, our data indicate that FBXW2 mediates SKP2 poly-ubiquitylation and subsequent targeted degradation and is a bona fide new E3 ligase for SKP2.

**Cell cycle-dependent regulation of β-TrCP1-FBXW2-SKP2 levels**. The above results showed that biochemically FBXW2 serves as a bridge to connect the two F-box proteins β-TrCP1 and SKP2, to form the β-TrCP1-FBXW2-SKP2 signalling cascade. We next determined under what physiological conditions the levels of these three F-box proteins are coordinately regulated. We chose to study cell cycle, given a promoting role of SKP2. As shown in Fig. 5a, when cells were arrested at the G0/G1 (via serum starvation), the level of FBXW2 was high, consistent with the high levels of tumour suppressors p21 and p27, which serve as positive controls. As cells entering the S phase (8 h post serum addition), the levels of VRK2 and β-TrCP1 started to increase, followed by FBXW2 decrease and SKP2 increase, and subsequent reduction of p21/p27 (at 8–12 h). These coordinated levels among three F-box proteins were maintained up to 32 h when a majority of cells were at the S or G2 phase. At 36 h when cells were reentering the G1, higher level of FBXW2 and low levels of β-TrCP1 and SKP2 was shown again. A cell cycle-dependent fluctuation of β-TrCP2 was similar to that of β-TrCP1 (Fig. 5a). Furthermore, silencing β-TrCP1 abrogates β-TrCP1-dependent fluctuation of FBXW2, leading to constitutively elevated levels of FBXW2 and substantially reduced levels of SKP2 (Supplementary Fig. 5a). We further confirmed that the levels of FBXW2 and SKP2 are negatively or positively regulated by β-TrCP1, as evidenced by FBXW2 reduction and SKP2 accumulation upon β-TrCP1 overexpression, and by FBXW2 accumulation and SKP2 reduction upon β-TrCP silencing, respectively (Supplementary Fig. 5b,c). Taken together, these results showed a cell cycle-dependent fluctuation of three F-box proteins in a coordinate manner with FBXW2 as a mediator of this F-box protein cascade by serving as a substrate of β-TrCP, and acting as an E3 for targeted degradation of SKP2.

**FBXW2 mediates the biological effects of β-TrCP1 and SKP2**. We then performed biological rescue experiment to determine whether this oncogene-tumour suppressor-oncogene axis coordinately regulates growth of lung cancer cells. We transfected β-TrCP1, FBXW2 (wt versus mutant) and SKP2 alone or in various combinations into lung cancer cells (Fig. 5b and

Supplementary Fig. 5d) and found that transfection of β-TrCP1 alone stimulated the monolayer growth and increased clonogenic survival of lung cancer cells. This effect can be blocked by simultaneous transfection of β-TrCP1 resistant FBXW2 mutant, FBXW2-3A (Fig. 5c,d and Supplementary Fig. 5e,f), suggesting that growth-stimulating effect of β-TrCP1 is mediated by targeted degradation of FBXW2. Similarly, increased monolayer growth and clonogenic survival trigged by ectopic expression of SKP2 could be rescued by simultaneous transfection of wild-type FBXW2 (Fig. 5e,f and Supplementary Fig. 5g,h). Furthermore, we silenced these three F-box proteins alone or in combinations (Fig. 5g and Supplementary Fig. 5i) and found that β-TrCP1 depletion resulted in FBXW2 accumulation to suppress cell growth and survival, which was reversed by simultaneous FBXW2 depletion (Supplementary Fig. 5i–k). Similarly, FBXW2 depletion caused SKP2 accumulation to stimulate cell growth and survival, which was abrogated by simultaneous SKP2 depletion both *in vitro* cell culture setting (Fig. 5g–i) and *in vivo* nude mice model (Fig. 5j,k). Note that tumour growth rate was the lowest in combinational group due to SKP2 depletion (Fig. 5j,k). Taken together, these results demonstrated a novel mechanism by which three F-box proteins (β-TrCP1-FBXW2-SKP2) coordinately regulate growth and survival of lung cancer cells by sequentially targeting each other for degradation.

**Correlation of three F-box protein levels with patient survival**. To further elucidate the cross-talk among the three F-box proteins in lung cancer, we measured their expression status in human lung cancer cell lines and tissues by immunoblotting (IB) and immuno-staining of a tumour tissue microarray (TMA). Among eight lung cancer cell lines and one immortalized line (BEAS-2B) tested, there is a general tendency of inversed protein levels between β-TrCP1 and FBXW2, as well as between FBXW2 and SKP2 (Fig. 6a), whereas the mRNA levels among them are similar in all lines (Supplementary Fig. 6a). We further tested their protein half-lives in A549 and H358 cells with opposite basal levels of three F-box proteins and found in general a positive correlation between longer protein half-life and higher basal levels, and vice versa (Supplementary Fig. 6b), suggesting post-translational regulation, particularly degradation rather than transcriptional regulation plays a key role for basal level difference. Furthermore, among 102 tumour TMA samples, the high FBXW2 staining group is largely associated with low staining group of β-TrCP1 ($P < 0.001$, $r = -0.392$, Pearson's $\chi^2$-test) and SKP2 ($P < 0.001$, $r = -0.478$, Pearson's $\chi^2$-test), whereas the low FBXW2 staining group is associated with high staining group of β-TrCP1 and SKP2 (Fig. 6b,c), respectively, further supporting a relationship in which FBXW2 is targeted by β-TrCP1, whereas SKP2 is targeted by FBXW2.

Given the overall patient survival data are available from these 102 tumour samples[58], we next correlated patient overall survival with expression of these three F-box proteins and found that expression of β-TrCP1 and SKP2 was positively, whereas expression of FBXW2 was negatively, associated with poor survival of lung cancer patients (Fig. 6d–f). To further extend above findings, we downloaded and analysed Affymetrix U133Plus2.0 microarray data including 226 lung adenocarcinomas and 20 normal lung tissues from Okayama *et al.*[59] 45 lung adenocarcinomas and 65 normal lung tissues from Hou *et al.*[60] and 172 lung adenocarcinomas from Zhu *et al.*[61]. We found that β-TrCP1 and SKP2 mRNA levels were always higher (Supplementary Fig. 6c,e and data not shown), whereas FBXW2 mRNA levels were lower in tumours than that in normal tissues (Supplementary Fig. 6d, and data now shown). Furthermore, patients with high mRNA levels of

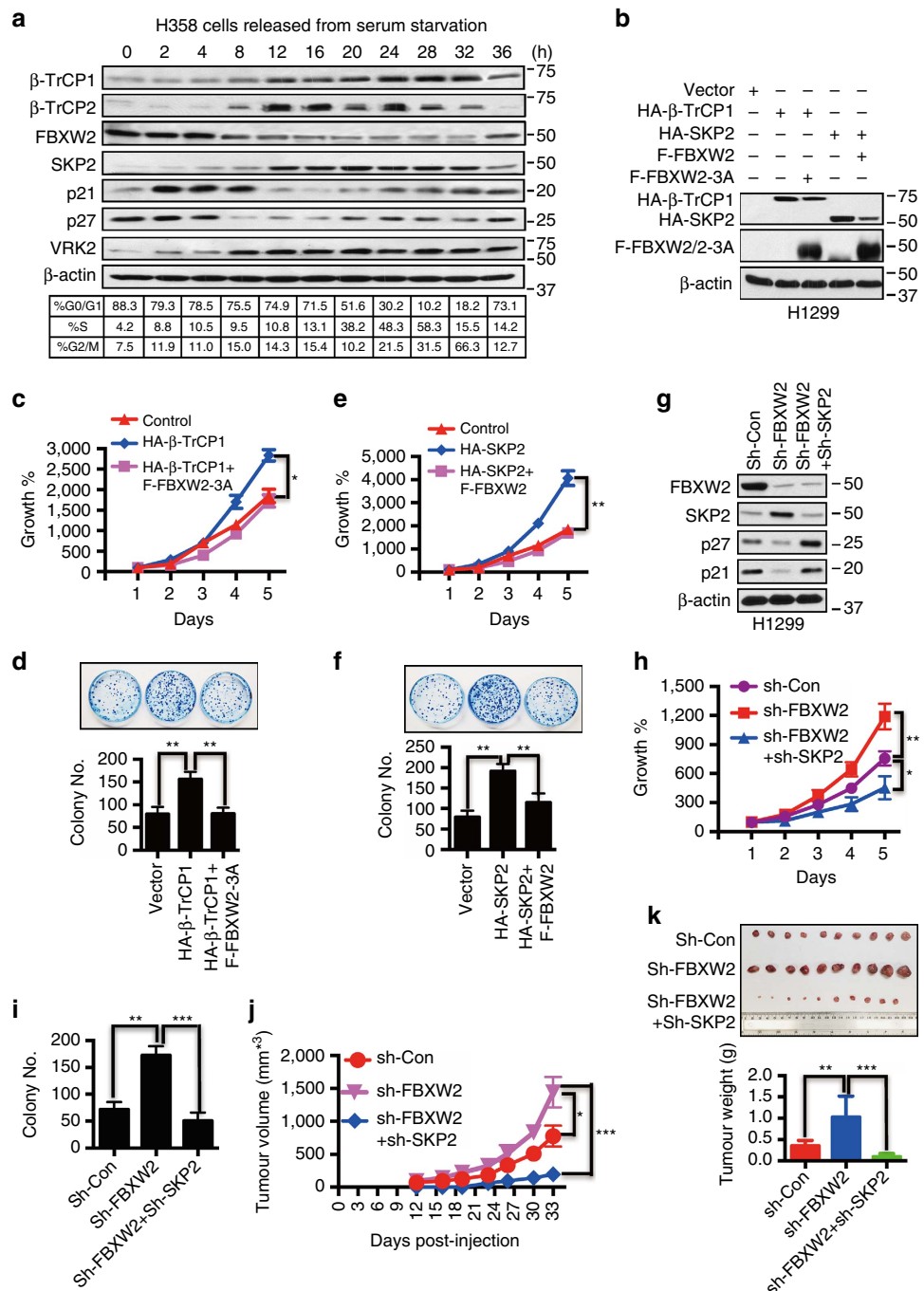

**Figure 5 | FBXW2 mediates the biological effects of β-TrCP1 as a downstream effector and of SKP2 as an upstream modulator. (a)** Fluctuation of the levels of F-box proteins and VRK2 kinase during cell cycle progression: H358 cells were serum starved for 48 h, followed by serum addition. Cells were harvested at indicated time points and subjected to FACS and IB analyses using indicated Abs. **(b–f)** FBXW2-3A mutant rescues growth-promoting phenotype induced by β-TrCP1 overexpression **(b–d)**, and wt FBXW2 rescues growth-promoting phenotype induced by SKP2 overexpression **(b,e,f)**: H1299 cells were co-transfected with the indicated plasmids, followed by IB **(b)**, ATP-lite proliferation assay ($n = 3$) **(c,e)** and clonogenic survival assay ($n = 3$) **(d,f)**. Shown is mean ± s.e.m. Student's $t$-test was performed, $*P < 0.05$; $**P < 0.01$. **(g–i)** FBXW2 depletion stimulates cell growth, which is abrogated by simultaneous SKP2 depletion: H1299 cells were transfected with shRNAs targeting FBXW2 alone or in combination with shRNA targeting SKP2, along with scramble control, and then harvested for IB **(g)**, ATP-lite proliferation assay ($n = 3$) **(h)** and clonogenic survival assay ($n = 3$) **(i)**. Shown is mean ± s.e.m. Student's $t$-test was performed, $*P < 0.05$; $**P < 0.01$; $***P < 0.001$. **(j,k)** FBXW2 depletion stimulates tumour growth, which is abrogated by simultaneous SKP2 depletion: H1299 cells were transfected with shRNAs targeting FBXW2 alone or in combination with shRNA targeting SKP2, along with scramble control, followed by injection ($1 \times 10^6$ cells) into nude mice. Tumour growth was observed for 33 days **(j)**. Tumours were then harvested, photographed and weighted **(k)**. Shown is mean ± s.e.m. Student's $t$-test was performed, $*P < 0.05$; $**P < 0.01$; $***P < 0.001$.

β-TrCP1 or SKP2 had a statistically significantly poorer survival rate, whereas those with low mRNA levels of β-TrCP1 or SKP2 had a better survival rate (Supplementary Fig. 6f,h and data not shown). In contrast, patients with high FBXW2 mRNA levels had a significantly better survival rate than those with low levels (Supplementary Fig. 6g and data not shown), further

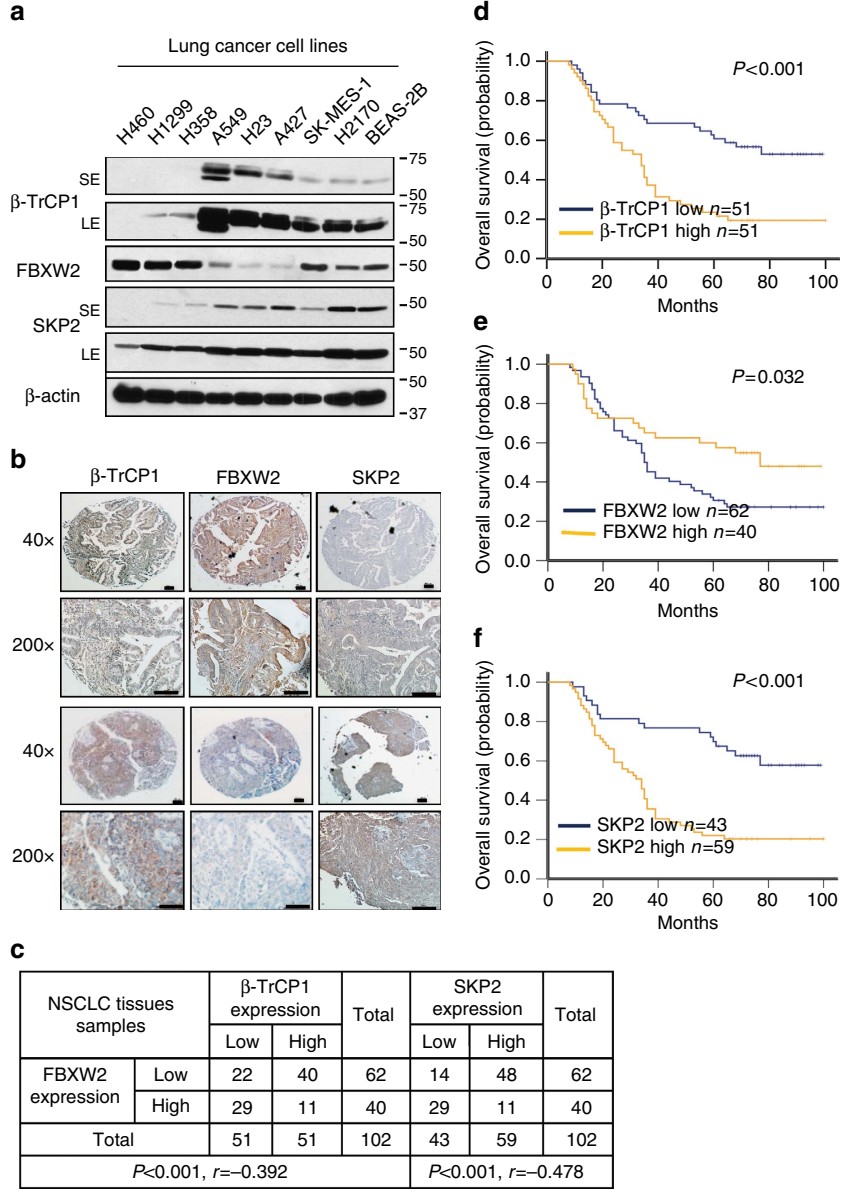

**Figure 6 | Expression of β-TrCP1/FBXW2/SKP2 in lung cancer cell lines and cancer tissues and their association with patient survival.**
(**a–c**) Expression levels among three F-box proteins in lung cancer cell lines and tissues: Cell lysates from eight lung cancer cell lines and one immortalized line (BEAS-2B) were subjected to IB (**a**); SE: Short exposure; LE: Long exposure. Lung cancer tissue microarrays were stained with indicated Abs and photographed (**b**, Scale bars, 100 μm), and data were then analysed using SPSS software to obtain coefficient (**c**; $P < 0.001$, Pearson's test). (**d–f**) Protein expression of three F-box proteins in lung cancer and their relationship with patient survival: Continuous protein expression values were classified into the low and high groups with equal number of patients, and 5-year survival time was used for Kaplan-Meier survival analysis (**d–f**). Kaplan-Meier survival analysis indicated that patient with higher expression of FBXW2 was related to a better overall survival (log-rank test, $P = 0.032$) (**e**); Higher expression of β-TrCP1 and SKP2 were related to a worse overall patient survival (log-rank test, $P < 0.001$ and 0.001, respectively) (**d,f**).

demonstrating that FBXW2 is a putative tumour suppressor in lung cancer.

**Gain or loss-of-function mutations of FBXW2 in human cancers.**
All the data shown so far strongly suggest that FBXW2 could be a tumour suppressor. To further determine whether FBXW2 is inactivated via mutations in human cancers, we searched the TCGA and COSMIC databases and found that FBXW2 is indeed mutated in various human cancers (Supplementary Fig. 7a). Although with very low frequency (~0.0001–0.001), we did find four germline mutations on V83I, S84C, I184V and

E419K (marked in yellow in Supplementary Fig. 7a) in general populations. We chose two mutants: FBXW2-S84C within the F-box domain and FBXW2-E269K within a WD40 substrate binding domain for functional characterization. Compared to wild-type FBXW2, mutant S84C had significantly reduced binding to Cul-1 and RBX1, but not SKP2, whereas mutant E269K had significantly reduced binding to SKP2, but not Cul-1 and RBX1, as expected (Fig. 7a). Biochemically, both FBXW2 mutants lost wild-type activity in shortening SKP2 half-life (Fig. 7b,c). Biologically, both mutants not only lost growth suppressive activity, but also gained, to various extents, the oncogenic activity, as evidenced by

their stimulation of monolayer growth and clonogenic survival, particularly in H1299 cells (Fig. 7d,e and Supplementary Fig. 7b,c). Furthermore, we characterized two FBXW2 mutants D30Y and G258W, found in lung cancer tissues with mutations localized outside of F-box and WD domain (Supplementary Fig. 7a), and found that they largely retained wild-type activity in suppression of cell growth (Supplementary Fig. 7d), indicating the importance of these two functional domains.

We next determined the biological function of the two FBXW2 mutants using an *in vivo* xenograft tumour model by inoculating subcutaneously the H1299 stable clones expressing vector, FBXW2-WT, S84C and E269K into both flank sides of nude mice. Compared to the vector control, wild-type FBXW2 significantly suppressed the *in vivo* tumour growth, whereas S84C mutant lost such a function, and E269K mutant gained oncogenic function by promoting *in vivo* tumour growth ($P < 0.01$, Fig. 7f). Consistently, the average tumour

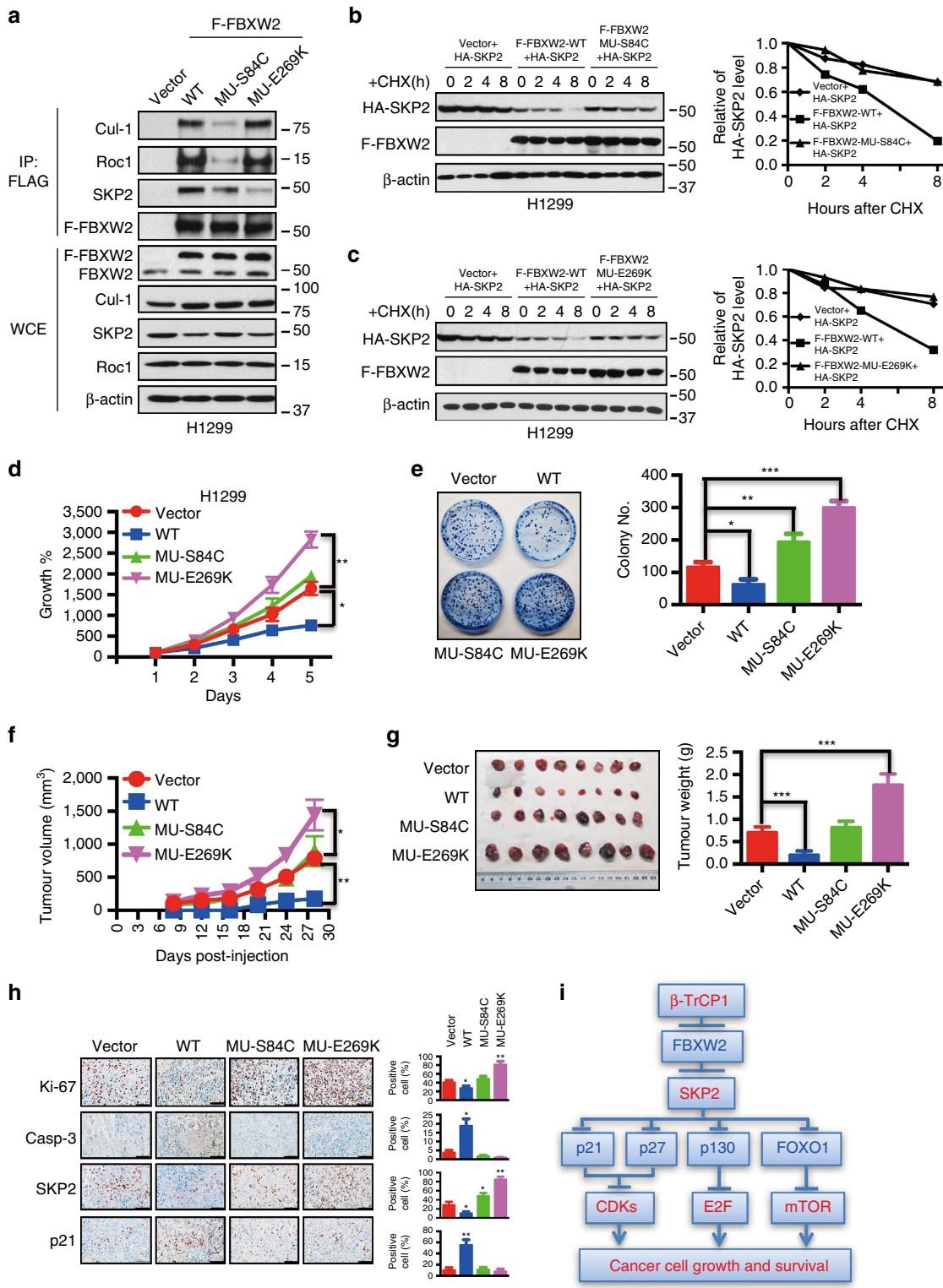

size and weight at the end of experiment (Day 28) were significantly higher in E269K mutant group ($P < 0.01$, Fig. 7g). Finally, immunohistochemical staining of tumour tissues revealed that E269K-expressing tumour had increased proliferation index (increased Ki-67 and decreased p21), reduced apoptosis index (cleavage caspase-3) and increased SKP2 expression, whereas wild-type FBXW2 had just an opposite effect (Fig. 7h). Taken together, the results derived from both *in vitro* cell culture setting and *in vivo* xenograft models coherently demonstrated that suppression of tumour growth by wild-type FBXW2 was lost in S84C mutant, whereas E269K mutant gained some oncogenic function to promote growth and survival of lung cancer cells. Thus, FBXW2 mutations found in human cancer can be either a loss-of-function or a gain-of-function mutation.

## Discussion

In the present study, we identified and characterized FBXW2, an orphan F-box protein[62] with unknown function in human cancer, as a physiological substrate of β-TrCP1. Our conclusion is supported by the following lines of evidence: (1) FBXW2 binds to β-TrCP1 under the physiological conditions; (2) FBXW2-β-TrCP1 binding is dependent on an evolutionarily conserved β-TrCP consensus binding motif (*SSGART*) on FBXW2; (3) Cellular levels of FBXW2 can be decreased or increased by β-TrCP1 overexpression or depletion, respectively; (4) FBXW2 protein half-life is shortened by wild-type β-TrCP1, but not by its F-box deletion mutant; (5) FBXW2-β-TrCP1 binding leads to FBXW2 ubiquitylation; (6) Given that FBXW2-β-TrCP1 binding is abrogated and FBXW2 protein half-life is extended by siRNA silencing or small molecule inhibition of CK1 and VRK2, our study also suggested that CK1 and VRK2 could be the kinases that phosphorylate FBXW2 at the β-TrCP1 binding motif to facilitate its β-TrCP1 binding and subsequent ubiquitylation and degradation. Consistently, CK1 is a kinase that phosphorylates serine residue at β-TrCP1 binding motif on Tiam1 (ref. 63), Yes-associated protein (ref. 64) and DEPTOR (ref. 28). Taken together, our results indicated that FBXW2 is a novel physiological substrate of the $SCF^{\beta-TrCP}$ E3 ubiquitin ligase.

Biological function of FBXW2 in any given human cancer is totally unknown. Here we showed that FBXW2 acts as a novel tumour suppressor in lung cancer with following lines of supporting evidence. (1) Ectopic expression of FBXW2 significantly inhibits the growth and survival of lung cancer cells; (2) SiRNA-based depletion of FBXW2 promotes the growth and survival of lung cancer cells; (3) FBXW2 is downregulated in lung cancer tissues and high levels of FBXW2 predicts a better patient survival; (4) FBXW2 is mutated in human cancers and FBXW2 mutants either loses tumour suppressor function or gains the oncogenic function; (5) Tumour suppressor function of FBXW2 is mediated by accumulation of the bona fide tumour suppressors, such as p21, p27, p130 and FOXO1, through targeted ubiquitylation and degradation of SKP2.

SKP2, a well-characterized oncogenic F-box protein, was first identified as a component of the cyclin A/CDK2 complex[65]. By inducing the degradation of the cyclin-dependent kinase inhibitors p21 and p27, the $SCF^{SKP2}$ E3 ubiquitin ligase promotes the S phase entry[66], whereas ubiquitylation and targeted degradation of SKP2 by the $APC^{CDH1}$ complex causes the stabilization of p27 or p21, leading to the G1/S phase arrest[56,57]. Here we showed that SKP2 is a novel substrate of FBXW2 with following lines of supporting evidence: (1) FBXW2 binds to SKP2 under physiological conditions; (2) FBXW2-SKP2 binding is dependent on an evolutionarily conserved FBXW2 putative consensus binding motif (*TSELLS*) on SKP2; (3) FBXW2 overexpression reduces SKP2 levels by shortening its protein half-life; (4) FBXW2 depletion increases SKP2 levels by extending its protein half-life; (5) FBXW2, but not its FBXW2ΔF mutant, promotes ubiquitylation and subsequent degradation of SKP2; (6) FBXW2 mediated negative regulation of SKP2 is independent of CDH1. At the present time, the only known substrate of FBXW2 is GCM1 (Glial cell missing homolog 1)[67,68], a transcription factor that regulate placental development[69,70] without any known implication in cancer. Identification of SKP2 as a novel substrate of FBXW2 extended the list of FBXW2 substrates and revealed its biological function in human cancer. We also postulated that FBXW2 consensus binding motif would likely be T/SSXXXS/T, awaiting for validation with other FBXW2 substrates yet-to-be identified and characterized.

By defining FBXW2 as a substrate of β-TrCP1 and an E3 for SKP2, we established, for the first time, a signalling cascade of three F-box proteins β-TrCP-FBXW2-SKP2 in lung cancer cells. Although F-box proteins, β-TrCP1 and SKP2 have been implicated in growth regulation of lung cancer cells[41,42,71] and many other human cancer cells[47], the cross-talking between them in coordinate regulation of cancer cell growth has never been previously reported. Based upon the biological functions of β-TrCP1, FBXW2 and SKP2 characterized in this study, we proposed an oncogene-tumour suppressor-oncogene axis in lung cancer cells. In support of this notion, we showed that (1) the levels of these three F-box protein is coordinately regulated during cell cycle progression with a high FBXW2 and low β-TrCP1 and SKP2 at the G0/G1 and vice versa at the S phase; (2) FBXW2 can mediate β-TrCP1 activity as a downstream effector and, on the other hand, manipulate SKP2 activity as an up-stream modulator to regulate growth and survival of lung cancer cells; (3) in lung cancer cells or tissues, there

**Figure 7 | FBXW2 mutants have the loss- or gain-of-function activity.** (**a**) FBXW2 mutants have reduced binding with SCF components: H1299 cells were transfected with indicated plasmids, followed by G418 selection. Resistant clones were pooled and subjected to FLAG-bead IP and IB with indicated Abs. (**b,c**) FBXW2 mutants MU-S84C (**b**) and MU-E269K (**c**) are unable to shorten SKP2 protein half-life: Stable H1299 cells were incubated with CHX for indicated time periods and harvested for IB. The band density was quantified. (**d,e**) Loss- or gain-of-function of FBXW2 mutants: H1299 cells stably expressing MU-S84C and MU-E269K mutants were subjected to ATP-lite proliferation assay ($n = 3$) (**d**), and clonogenic survival assay ($n = 3$) (**e**). Shown is mean ± s.e.m. Student's *t*-test was performed, $*P < 0.05$; $**P < 0.01$; $***P < 0.001$. (**f,g**) FBXW2 mutants either lose the tumour suppressor function or gain the oncogenic function *in vivo*: H1299 cells stably expressing MU-S84C and MU-E269K mutants ($1 \times 10^6$ cells) were inoculated s.c. in both flanks of nude mice. The tumour growth was monitored twice a week for up to 28 days and growth curve plotted (**f**). Tumour tissues were harvested, photographed and weighed at 28 days (**g**). Student's *t*-test was used to compare each experimental group with the control group. Shown are mean ± s.e.m., $*P < 0.05$; $**P < 0.01$; $***P < 0.001$. (**h**) Immunohistochemical staining of xenograft tumour tissues. Tumour tissues from four groups of mice were fixed, sectioned and stained with indicated antibodies. Scale bars: 100 μm. Shown are mean ± s.e.m. $*P < 0.05$; $**P < 0.01$. (**i**) The oncogene-tumour suppressor-oncogene axis—a working model: During tumorigenesis, oncogenic β-TrCP1 is activated[80] to promote ubiquitylation and degradation of FBXW2 tumour suppressor, resulting in abrogation of FBXW2-induced SKP2 degradation. Accumulated oncogenic SKP2 promotes ubiquitylation and degradation of tumour suppressors such as p21, p27, p130, FOXO1, and leading to activation of CDKs, E2F and mTOR, eventually to uncontrolled proliferation of cancer cells.

is a general tendency in which high levels of β-TrCP1 are paired with low levels of FBXW2, whereas high levels of FBXW2 are paired with low levels of SKP2, and vice versa; (4) at the levels of both protein and mRNA, lung cancer patients with high expression of β-TrCP1 or SKP2 had a significantly poorer survival, whereas higher expression of FBXW2 had a better survival. It is worth noting, however, that inverse correlation at the protein levels of β-TrCP1 versus FBXW2, and FBXW2 versus SKP2 are likely through targeted degradation of downstream substrates, whereas correlated changes at the mRNA levels must occur via yet-to-be defined mechanisms at the genome and/or transcriptional levels during lung tumorigenesis, a subject warrant future investigation. Nevertheless, our data support the notion that this oncogene-tumour suppressor-oncogene axis may operate during human lung tumorigenesis. Targeted inactivation of β-TrCP1 or SKP2 (ref. 72) could, therefore, have a therapeutic value for the treatment of lung cancer.

Inactivation of tumour suppressive F-box protein FBXW7 by point mutation or deletion was frequently found in a variety of human cancers[45]. Through TCGA database search, we found numerous FBXW2 mutations in human cancers including lung cancer. Functional characterization of two selected mutations showed that F-box domain mutant (S84C) is mainly a loss-of-function mutant, whereas WD40 domain mutant (E269K) has a gain-of-function property. The loss of wild-type FBXW2 function could be attributable to their inability of degrading SKP2. It is also likely that gain of oncogenic function of E269K mutant involves the binding and degrading other yet-to-be identified tumour suppressor protein(s). Future study is directed to define the underlying mechanism.

In summary, our study reveals an important interplay among F-box proteins, namely β-TrCP1, FBXW2 and SKP2. The β-TrCP-FBXW2-SKP2 forms an oncogene-tumour suppressor-oncogene axis to regulate cell cycle progression, proliferation and survival of lung cancer cells by keeping the levels of each other in check. When cells were arrested in response to serum starvation, the level of tumour suppressive FBXW2 is high to keep SKP2 level low and p21/p27 high. Upon serum addition, VRK2 level increases to phosphorylate FBXW2 and facilitate its binding to increased β-TrCP for targeted degradation. FBXW2 reduction results in SKP2 accumulation to promote targeted ubiquitylation and degradation of a number of tumour suppressors and apoptosis-inducing substrates such as p21, p27, p130 and FOXO1, leading to the activation of the CDKs, E2F and mTOR pathway[73] to promote cell proliferation and survival, eventually tumour formation. FBXW2 inactivation via point mutations (for example, S84C and E269K) would abrogate SKP2 degradation, thus promoting tumorigenesis (Fig. 7i).

## Methods

**Cell culture.** All the cell lines used in this study were obtained from ATCC and cultured at 10% Dulbecco's modified Eagle's medium. Cell lines were frequently examined to ensure free of mycoplasma contamination.

**Immunoblotting and immunoprecipitation.** For direct IB analysis, cells were lysed in a Triton X-100 or RIPA buffer with phosphatase inhibitors. The following primary antibodies were used: goat-FBXW2 (sc-160326, SANTA CRUZ; 1:1,000 overnight, 4 °C); rabbit-FBXW2 (ab83467, Abcam; 1:1,000 overnight, 4 °C); rabbit-FBXW2 (#11499-1-AP, Proteintech; 1:500 overnight, 4 °C); rabbit-SKP2 (#2652, Cellsignal; 1:1,000 overnight, 4 °C); rabbit-β-TrCP1 (D13F10) (#4394, cellsignal; 1:1,000 overnight, 4 °C); rat-HA (#11867423001, Roche Life Science; 1:2,000 overnight, 4 °C); mouse-FLAG (#F1804, Sigma; 1:2,000 overnight, 4 °C); mouse-P21 (#556430, BD Pharmingen; 1:2,000 overnight, 4 °C); mouse-P27 (#554069, BD Pharmingen; 1:2,000 overnight, 4 °C); mouse-β-Actin (sc-47778, SANTA CRUZ; 1:10,000 overnight, 4 °C); rabbit-Cleaved Notch1 (Val1744) (#4147, Cellsignal; 1:1,000 overnight, 4 °C); rabbit-GRK2

(C-15) (sc-562, SANTA CRUZ; 1:1,000 overnight, 4 °C); rabbit-VRK2 (H-255)(sc-98733, SANTA CRUZ; 1:1,000 overnight, 4 °C); rabbit-FBXL11 (H-120)(sc-135126, SANTA CRUZ; 1:1,000 overnight, 4 °C); rabbit-FBXL3 (GTX110755, GeneTex; 1:1,000 overnight, 4 °C); mouse-p130 (A-10) (sc-374521, SANTA CRUZ; 1:1,000 overnight, 4 °C); rabbit-FBXW11 (#13149-1-AP, Proteintech; 1:1,000 overnight, 4 °C); rabbit-wee1 (#4936, Cell-signal; 1:1,000 overnight, 4 °C); rabbit-c-jun (sc-1694, SANTA CRUZ; 1:1,000 overnight, 4 °C); rabbit-cul-1(sc-11384, SANTA CRUZ; 1:1,000 overnight, 4 °C); rabbit-FOXO1 (#2880, Cellsignal; 1:1,000 overnight, 4 °C); mouse-cdh1 (#CC43, Calbiochem; 1:1,000 overnight, 4 °C); rabbit-rbx1 (made by our lab; 1:1,000 overnight, 4 °C).

To immunoprecipitate endogenous proteins, whole cell extracts were pre-cleared with normal IgG-AC (Santa Cruz) followed by overnight incubation at 4 °C with antibody against FBXW2, β-TrCP, FBXL3, FBXL11 or SKP2; to immunoprecipitate exogenously expressed FLAG-tagged or HA-tagged proteins, the pre-cleared cell lysates were incubated with bead-conjugated FLAG (Sigma) or with HA antibody (Roche) in a rotating incubator overnight at 4 °C. The immunoprecipitates were washed with lysis buffer and assessed by IB.

All the original X-films for IB analysis presented in this study are shown in Supplementary Fig. 8.

**Half-life analysis of FBXW2 and SKP2.** Cells were transiently transfected with different plasmids or siRNA against FBXW2 for 48 h, and then treated with 20 μg ml$^{-1}$ cycloheximide, followed by collection of transfected cells at indicted time points for IB assay. The relative FBXW2 or SKP2 levels were quantified by densitometry analysis using the ImageJ1.410 image processing software.

**Lentivirus or siRNA transfection.** FBXW2 and GRK2 siRNAs are pools including 2–3 target-specific 19–25 nt siRNAs from Santa Cruz. Other siRNA or shRNA targeting sequences are as follows. si-VRK2: 5′-GCAAGGUUCUGGAU GAUAUUU-3′; si-β-TrCP: 5′-AAGTGGAAT TTGTGGAACATC-3′; si-SKP2: 5′-GCUUCACGUGGGGAUGGGAdTdT-3′; si-CDH1: 5′-AATGAGAAGT CTCCC AGTCAGdTdT-3′ and si-Con: 5′-AUUGUAU GCGAUCGCAGA CUU-3′. sh-β-TrCP1: 5′-GCGTTGTATTCGATTTGATAA-3′, sh-FBXW2: 5′-CCGGGGCCTTTGAAACCTCGTCA TTACTCGAGTAATGACGAG GTTTC AAAGGCTTTTTG-3′ and sh-SKP2: 5′-AAGGT CTCTGGTGTTTGTAA G-3′ (ref. 74). The lentiviruses expressing these sh-RNAs were prepared at the University of Michigan Vector Core. For gene silencing, cells were transfected with lentivirus-based shRNA or oligonucleotides targeting various genes in Dulbecco's modified Eagle's medium or RPMI-1640 with 10 × Lentivirus or 200 nM siRNAs using DharmaFECT transfection reagent (GE Healthcare) according to the manufacturer's protocol.

**Quantitative real-time PCR.** RNA was isolated using RNeasy reagents (Qiagen, Hilden, Germany) and then transcribed into cDNA using SuperScript III reagents (Invitrogen) with an oligo(dT)$_{20}$ primer. Quantitative real-time PCR was performed using Power SYBR Green Mastermix (Applied Biosystems, Foster City, CA, USA) on an Applied Biosystems 7900HT Real-Time PCR System. All oligonucleotide primers were obtained from Integrated DNA Technologies (Coralville, IA, USA). The housekeeping gene, GAPDH, was used as loading controls. The sequences of primer sets were 5′-GGACACAAAC GAGGCATTGCCT-3′ and 5′- CAACGCACCAATTCCTCATGGC-3′ for β-TrCP1; 5′-GTTCAGCACTTGGTCTCTACCAG-3′ and 5′-GTAGCGGTTGT CAAACAGCA GG-3′ for FBXW2; 5′-GATGTGACTGGTCGGTTGCTGT-3′ and 5′-GAGTTCGATAGGTCC ATGTGCTG-3′ for SKP2; 5′-GTCTCCT CTGACTTCAACAGCG-3′ and 5′-ACCACCCTGTT GCTGTAGCCAA-3′ for GAPDH.

**The in vivo ubiquitylation.** To determine the ubiquitylation of FBXW2 or FBXW2-3A mutant by β-TrCP1, H1299 or 293 cells were co-transfected with β-TrCP1, His-HA-Ub, and FBXW2 or FBXW2-3A, along with empty vector or β-TrCP1ΔF controls. To determine SKP2 ubiquitylation by FBXW2, H1299 or 293 cells were co-transfected with FBXW2, SKP2, and His-HA-Ub, along with empty vector or FBXW2ΔF controls. To determine SKP2 ubiquitylation by other F-box proteins, 293 cells were co-transfected with FBXW2, FBXW4, FBXW8, SKP2 and His-HA-Ub, along with empty vector control. Cells were lysed in 6 M guanidinium denaturing solution, as described previously[75]. FBXW2-poly-Ub or SKP2-poly-Ub was purified by Ni-bead pull-down and detected by IB using anti-FBXW2 or anti-FLAG Ab.

**The in vitro ubiquitylation.** HA-β-TrCP1 or HA-β-TrCP1ΔF was precipitated from 293 cells transfected by either protein and eluted with 3 × HA peptide (Sigma), serving as the source of E3; FLAG-tagged FBXW2 or FBXW2-3A was pulled down by FLAG bead (Sigma) after transfection into 293 cells, serving as substrate. For SKP2 ubiquitylation by FBXW2, HA-FBXW2 or HA-FBXW2ΔF E3 was precipitated from 293 cells transfected by either protein and eluted with 3 × HA peptide (Sigma); FLAG-tagged SKP2 was pulled down by FLAG bead

after transfection into 293 cells and used as the substrate. The reaction was carried out at 30 °C for 1 h in 30 µl reaction buffer (40 mM Tris-HCl, pH 7.5, 2 mM DTT, 5 mM MgCl2) in the presence of E1, E2 and purified E3s. Poly-ubiquitinated FBXW2 or SKP2 was resolved by SDS–polyacrylamide gel electrophoresis and detected by IB with anti-FLAG Ab.

**ATPlite cell proliferation, clonogenic and soft agar assays.** Cells were seeded into 96-well plates or 60-mm dishes, followed by ATPlite cell proliferation assay (Perkin-Elmer) for cell proliferation at different time, for clonogenic and soft agar assays with colonies stained and counted after incubation for 12–15 days[76,77].

**FACS analysis.** Cells were synchronized at G0/G1 by serum starvation for 48 h and then released by serum addition. Cells were then collected at the indicated time points. Single-cell suspensions was fixed in ice-cold 70% ethanol overnight, labelled with 500 µl propidium iodide (50 µl ml$^{-1}$; Sigma-Aldrich) for at least 15 min in dark at room temperature, and analysed directly on an LSR II flow cytometer (BD Biosciences, San Jose, CA, USA).

**Immunohistochemical staining.** The human lung cancer tissue arrays were from the Sun Yat-Sen University, Guanzhou, China. Immunohistochemical staining of human lung cancer TMAs or mice tumours was performed as described previously[78]. Briefly, after deparaffinization, rehydration, antigen retrieval and blocking, the tissue slides were incubated overnight at 4 °C with indicated antibodies. The following primary antibodies were used: rabbit-FBXW2 (ab83467, Abcam; 1:500 overnight, 4 °C); rabbit-SKP2 (#2652, Cellsignal; 1:1,000 overnight, 4 °C); rabbit-β-TrCP1 (D13F10) (#4394, Cellsignal; 1:500 overnight, 4 °C); mouse-P21 (#556430, BD Pharmingen; 1:1,000 overnight, 4 °C); mouse-ki-67 (#550609, BD Pharmingen; 1:1,000 overnight, 4 °C); rabbit-Cleaved Caspase-3 (Asp175) (#9661, Cellsignal; 1:500 overnight, 4 °C).

**Database mining.** The Cancer Genome Atlas (TCGA) data portal (https://tcga-data.nci.nih.gov/tcga/) and the Catalogue of somatic mutations in cancer (Cosmic, v74) (http://cancer.sanger.ac.uk/cosmic/) were used to identify somatic mutations in FBXW2 (missense, nonsense, splice and frame-shift indel) in all tissue types. The FBXW2 somatic mutations from the above two database were integrated and combined and a consensus list of FBXW2 mutations were presented in Supplementary Fig. 7a.

**Animal experiments.** Four- to six-week-old BALB/c athymic nude mice (nu/nu, female) were used with each experimental group consisting of five mice. All animal experiments were carried out according to a protocol approved by the University of Michigan, Committee for Use and Care of Animals. $1 \times 10^6$ H1299 stable cell lines (Vector; FLAG-FBXW2-WT; FLAG-FBXW2-MU-S84C; FLAG-FBXW2-MU-E269K; Sh-FBXW2; Sh-FBXW2/SKP2) were mixed 1:1 with matrigel (BD biosciences) in a total volume of 0.2 ml and were injected subcutaneously into both flanks of mice The growth of tumours was measured twice a week and average tumour volume (TV) was calculated according to the equation: $TV = (L \times W^2)/2$.

**Patients and clinical specimens.** The present study included 102 patients with NSCLC, diagnosed at the Cancer Center, Sun Yat-sen University between 2002 and 2003 (ref. 58). The patients, 73 males (71.6%) and 29 females (28.4%), ranged in age from 33 to 84 years. Histological examination was performed on formalin-fixed tissues in all cases and tumours were diagnosed and classified according to the AJCC (American Joint Committee on Cancer guidelines) classification. All the patients (from stage I to stage III) underwent radical surgery of primary tumour and lymph nodes, and received adjuvant cisplatin-based chemotherapy after surgery. No chemotherapy or radiotherapy was given to patients before surgery. Written informed consent for the use of the tissues was obtained from all patients before surgery, and the study was approved by the Institute Research Ethics Committee of Sun Yat-sen University.

**Primary tumour-derived gene expression data sets.** Three published Affymetrix microarray data sets[59–61] were used in the survival or tumour versus normal comparison analyses for FBXW2, β-TrCP1 and SKP2. The CEL files of microarray data were normalized using Robust Multi-array Average (RMA) method[79], and then log 2 transformed.

**Statistical analyses.** Overall survival was the outcome for lung cancer patients (corresponding to TMA set) as well as Okayama and Zhu's data sets, and it was censored at 5 years. Survival functions were plotted using Kaplan-Meier method, and comparison of survival functions was performed by the log-rank test. The Student's t-test was used for comparing marker expression or other parameters between two groups (for example, lung tumour and normal lung tissues). Boxplot was used to show the difference between tumour and normal. Pearson

correlation was used to analyse the association between proteins expression on tumour tissues. The P value less than 0.05 was considered as statistically significant. Statistical Program for Social Sciences software 20.0 (SPSS, Chicago, IL, USA) was used. All statistical tests were two-sided.

**Data availability.** The Cancer Genome Atlas (TCGA) data referenced during the study are available in a public repository from the https://cancergenome.-nih.gov/ website. The Catalogue Of Somatic Mutations In Cancer (COSMIC, v74) data referenced during the study are available in a public repository from the http://cancer.sanger.ac.uk/cosmic website. The authors declare that all the other data supporting the findings of this study are available within the article and its Supplementary Information files and from the corresponding author on reasonable request.

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

## Acknowledgements

We thank Dr M. Pagano for generously providing plasmids encoding F-box proteins. We also thank Dr Q. Liu to give us the permission to use tumour tissue microarrays from lung cancer patients. This work is supported by NIH Grants R01CA156744 and CA171277 (Y.S.); by the Chinese NSFC Grants 81572718 (Y.S.), and by the Chinese Minister of Science and Technology grant 2016YFA0501800 (Y.S.). The author (Y.S.) also gratefully acknowledges the support of K.C. Wong Education Fund.

## Author contributions

J.X. and Y.S. conceived and designed the study. J.X., W.Z., F.Y., G.C., H.L., Y.Z., P.L., H.L., M.T. and X.X. designed and performed experiments, and analysed data. J.X., W.Z., and Y.S. wrote the manuscript.

## Additional information

**Competing financial interests:** The authors declare no competing financial interests.

