## [Peer Review File · Nature Communications]

Reviewers' comments:

Reviewer #1 (F-box proteins)
(Remarks to the Author):

The manuscript by Xu et al. suggests an intriguing hypothesis on a β TrCP-FBXW2-SKP2 signaling cascade. In this model, FBXW2, a relatively underexplored F-box protein, is postulated to transmit β TrCP signal to SKP2 pathway. Specifically, the authors have provided a large set of data indicating that FBXW2 is a previously unrecognized substrate of SCF β TrCP E3, but SCF FBXW2 acts as an E3 to mediate the ubiquitination and degradation of SKP2, thereby diminishing the SCF SKP2 E3 activity. In addition, there appears to be evidence that FBXW2 may have a tumor suppressor role. Overall, this paper is significant by providing a new proteolytic signaling mechanism that may impact tumorigenesis.

However, it is not clear to this reviewer how the β TrCP-FBXW2-SKP2 signaling cascade operates because the role of VRK2 kinase has not been clearly defined. First, while the expression of VRK2 was serum-dependent, the kinase level remained constant in the progression of cell cycle (Fig. 5A). Does it mean that the FBXW2 degron SSGART remains constitutively phosphorylated and that the key regulation is the relative abundance of β TrCP1, whose expression is cell cycle regulated? What about β TrCP2? Second, is there any evidence demonstrating that VRK2 directly phosphorylates the FBXW2 degron SSGART? The paper will be strengthened significantly if the authors could perform peptide experiments showing that the phosphor-peptide, but not the unmodified form, binds to β TrCP1 directly.

Minor comments:

Fig. 1A: please define "input" by percentage.

Please provide quantification for Fig. 2a, c, d and e.

Please improve clarify for the following statements: 1. "Furthermore, the levels of β TrCP1-FBXW2-SKP2 in tumor tissues are inversely correlated with FBXW2 predicting a better, whereas β -TrCP1 and SKP2 a worse patient survival." 2. "Likewise, ectopically expressed β -TrCP1 failed to pull-down FBXW2-3A (Supplementary Fig. 2b), indicating targeted degradation is the binding dependent." 3. "As shown in Fig. 5a, when cells were arrested at the G0/G1 (via serum starvation), the levels of tumor suppressive FBXW2, as well as p21 and p27 were high."

Reviewer #2 (ubiquitination and cancer)
(Remarks to the Author):

In this manuscript, the authors describe the β TrCP-FBXW2-SKP2 axis that is a newly identified SCF E3 cascade in lung cancer. The authors indicated that FBXW2, which is a less characterized orphan F-box protein, is targeted by β TrCP for ubiquitination and subsequent degradation. Additionally, they found FBXW2 directs poly-ubiquitination of SKP2, an oncogenic F-box protein, to facilitate its proteasome-dependent degradation. The expression levels of both β TrCP and SKP2 are significantly elevated in lung cancer clinical samples, which are inversely correlated with FBXW2 expression. These observations imply that FBXW2 serves as a tumor suppressor in lung cancer. In line with this finding, the forced expression of cancer-derived FBXW2 mutants promoted tumor growth in xenograft experiments. Most of the assays are well-controlled and carried out using standard techniques that are well-established. However, there are a few more experiments that would need to be appropriately

performed to confirm that the proposed F-box protein axis is functional in lung cancer cells.

Specific comments:

Is this degradation cascade a cell-type specific or ubiquitous mechanism? The author should clarify why they specifically look at lung cancer model.

Figure 1i and 4i: The data presented in these panels are not quantitative. The authors should employ real-time qPCR. In addition, the qPCR assays should be performed using the RNA samples derived from the cells analyzed in Fig 1e, Fig 4e, and Fig 6a to quantify mRNA levels of betaTrCP, FBXW2 or SKP2 where required.

Figure 1h: The CHX assay does not seem to be working properly. This experiment needs to be repeated.

Figure 2d-2e: Endogenous FBXW2 should be analyzed.

Identification of the association between FBXW2 and SKP2 seems to be somewhat abrupt as it was identified through prediction based on the WB data of FBXW2 manipulated cells. Although the FBXW4 and FBXW8 are incorporated as a negative control in Supplementary Figure 4i-4j, it may be better to perform co-IP experiments using a panel of F-box proteins in order to demonstrate the specificity of the FBXW2-SKP2 interaction.

Figure 5a: The synchronization experiment should be performed using control- and betaTrCP-shRNA cell lines side by side to investigate how betaTrCP depletion affects the expression of FBXW2 and SKP2, through different cell cycle phases.

Figure 5: To validate an FBXW2 tumor suppressor function, a xenograft experiment should be carried out using a set of following cell lines: sh-Con, shFBXW2, and shFBXW2+shSkp2.

Figure 6a: The FBXW2 protein half-life should be analyzed and compared among multiple betaTrCP high- and low-expressing lung cancer cell lines to determine if the varying expression levels of FBXW2 is due to its altered half-life. Likewise, the half-life of SKP2 should be measured, but instead using multiple FBXW2 high- and low-expressing lung cancer cell lines.

Reviewer #3 (Lung cancer and fbw proteins)
(Remarks to the Author):

This manuscript outlines molecular interaction between three SCF-based F-box proteins in cancer. The authors indicate that a relatively new F-box protein, FBXW2, partakes within a Beta-TrCP-FBXW2-SKP2 axis that modulates lung cancer cell growth. The new contribution here really is the intermediary role of FBXW2 in cancer, as the roles of the other two F-box proteins in neoplasia (and lung cancer) are well characterized. The data presented incorporates biochemical interaction studies of the effectors, studies in lung tissue, and in vivo experiments. Some data as presented are not compelling for direct molecular interactions between F box proteins and several controls lacking.

- 1) The authors have not mapped the binding motifs between FBXW2 and SKP2.
- 2) The authors suggest that BetaTrCP targets FBXW2 for polyubiquitination. The data are not compelling. First, there is no demonstration that endogenous FBXW2 is polyubiquitinated by the E3

complex. Second, the identification of an acceptor site is not demonstrated either withn SKP2 or FBXW2. The type of polyubiquitination (K63/K48) is not identified.

3) The SCF complexes recognize phosphodegrons and identity of a phospho-site within SKP2 that modulates interaction between FBXW2 as a binding partner and therefore polyubiquitination is not known.

Other concerns.

1) many Co-IPs lack controls, mention of % input etc. For example, Fig 1a another F-box subunit as a specificity controlis lacking, a control sample where FBXW2 is depleted from cells before IP is not shown.

2) Fig 1 h is not compelling. There is a 90% depletion of Beta-TrCP by knockdown and yet only a tiny increase in FBXW2.

3) Some panels in Fig 3 again would have been more convincing with inclusion of another F-box as a specificity control.

4) Figs 6b and 7h lack scale bars

Dear Reviewers,

Thank you for your positive comments on our manuscript and your thoughtful critiques. Below, we have addressed point-by-point your constructive critiques, and modified the text accordingly.

Responses to Reviewer #1: (Expert in F-box proteins)

The manuscript by Xu et al. suggests an intriguing hypothesis on a β TrCP-FBXW2-SKP2 signaling cascade. In this model, FBXW2, a relatively underexplored F-box protein, is postulated to transmit β TrCP signal to SKP2 pathway. Specifically, the authors have provided a large set of data indicating that FBXW2 is a previously unrecognized substrate of SCF β TrCP E3, but SCF FBXW2 acts as an E3 to mediate the ubiquitination and degradation of SKP2, thereby diminishing the SCF SKP2 E3 activity. In addition, there appears to be evidence that FBXW2 may have a tumor suppressor role. Overall, this paper is significant by providing a new proteolytic signaling mechanism that may impact tumorigenesis.

We thank this reviewer for his/her positive comments on our manuscript.

However, it is not clear to this reviewer how the β TrCP-FBXW2-SKP2 signaling cascade operates because the role of VRK2 kinase has not been clearly defined. First, while the expression of VRK2 was serum-dependent, the kinase level remained constant in the progression of cell cycle (Fig. 5A). Does it mean that the FBXW2 degron SSGART remains constitutively phosphorylated and that the key regulation is the relative abundance of β TrCP1, whose expression is cell cycle regulated? What about β TrCP2? Second, is there any evidence demonstrating that VRK2 directly phosphorylates the FBXW2 degron SSGART? The paper will be strengthened significantly if the authors could perform peptide experiments showing that the phosphor-peptide, but not the unmodified form, binds to β TrCP1 directly.

We value reviewer's concern. Indeed, we found that VRK2 levels increased upon serum addition, and elevated levels after 8 hrs remained throughout the rest of cell cycle. Our data did suggest after 8 hrs of serum addition that "the FBXW2 degron SSGART remains constitutively

phosphorylated and that the key regulation is the relatively abundance of β TrCP1, whose expression is cell cycle regulated”, as correctly pointed out by the reviewer. We did see a perfect inverse correlation in the levels between FBXW2 and β TrCP1 (Fig. 5a).

Also, as reviewer suggested, we measured the levels of β TrCP2 during cell cycle and found it fluctuated in consistent with β TrCP1 and in an inverse correlation with FBXW2 (Figure 5a).

Finally, following reviewer’s suggestion, we performed peptide experiments and showed that the phosphor-FBXW2 peptide, but not the unmodified form, binds to β TrCP1 directly. This newly generated data is now included as Figure 1b.

Minor comments:

Fig. 1A: please define "input" by percentage.

We have defined “input” as 2%.

Please provide quantification for Fig. 2a, c, d and e.

Per reviewer’s suggestion, we have quantified these data and included the relative levels in these figures.

Please improve clarify for the following statements:

1. *"Furthermore, the levels of β TrCP1-FBXW2-SKP2 in tumor tissues are inversely correlated with FBXW2 predicting a better, whereas β -TrCP1 and SKP2 a worse patient survival."*

We have modified the statement and it now reads “The level of FBXW2 is inversely correlated with that of β TrCP1 and SKP2 with FBXW2 predicting a better, whereas β -TrCP1 and SKP2 a worse patient survival.” (page 2)

2. *"Likewise, ectopically expressed β -TrCP1 failed to pull-down FBXW2-3A (Supplementary Fig. 2b), indicating targeted degradation is the binding dependent."*

We have modified the statement and it now reads “Likewise, ectopically expressed β -TrCP1 failed to pull-down FBXW2 mutant, FBXW2-3A, indicating the degradation of FBXW2 by β -TrCP1 is binding dependent” (page 8)

3. *"As shown in Fig. 5a, when cells were arrested at the G0/G1 (via serum starvation), the levels of tumor suppressive FBXW2, as well as p21 and p27 were high."*

We have modified the statement and it now reads “As shown in Fig. 5a, when cells were arrested at the G0/G1 (via serum starvation), the level of FBXW2 was high, consistent with the high levels of tumor suppressors p21 and p27, which serve as positive controls.” (page 13)

Responses to Reviewer#2 (Expert in ubiquitination and cancer)

In this manuscript, the authors describe the betaTrCP-FBXW2-SKP2 axis that is a newly identified SCF E3 cascade in lung cancer. The authors indicated that FBXW2, which is a less

characterized orphan F-box protein, is targeted by betaTrCP for ubiquitination and subsequent degradation. Additionally, they found FBXW2 directs poly-ubiquitination of SKP2, an oncogenic F-box protein, to facilitate its proteasome-dependent degradation. The expression levels of both betaTrCP and SKP2 are significantly elevated in lung cancer clinical samples, which are inversely correlated with FBXW2 expression. These observations imply that FBXW2 serves as a tumor suppressor in lung cancer. In line with this finding, the forced expression of cancer-derived FBXW2 mutants promoted tumor growth in xenograft experiments. Most of the assays are well-controlled and carried out using standard techniques that are well-established. However, there are a few more experiments that would need to be appropriately performed to confirm that the proposed F-box protein axis is functional in lung cancer cells.

We thank this reviewer for his/her positive comments on our work.

Specific comments:

Is this degradation cascade a cell-type specific or ubiquitous mechanism? The author should clarify why they specifically look at lung cancer model.

We value reviewer's concern. We believe that this degradation cascade is a ubiquitous mechanism, since the ubiquitylation and degradation cascade can also be detected in 293 kidney cells (Fig. 2f), in addition to multiple lung cancer lines. The reasons we focused on lung cancer cells are as follows:

- 1) Abnormality of SKP2 was frequently reported in lung cancer. We have now added few related references (refs. 44-47) in the Introduction section (page 4);**
- 2) We did find FBXW2 mutations in lung cancer tissues through TCGA database mining (Supplementary Fig. 7a);**
- 3) Our laboratory has recently focused our attention to lung cancer models which harbors activated SCF-CRL ligase: [1] JCI, 124:835-846, 2014; 2) JNCI, 2014 May 22; 106(6): dju083. doi: 10.1093/jnci/dju083; 3) Oncotarget, 5:6746-6755, 2014; 4) Clinical Can Res, 2016 Sep 2. pii: clincanres.1585.2016. PMID:27591266; DOI:10.1158/1078-0432.CCR-16-1585].**

Figure 1i and 4i: The data presented in these panels are not quantitative. The authors should employ real-time qPCR. In addition, the qPCR assays should be performed using the RNA samples derived from the cells analyzed in Fig 1e, Fig 4e, and Fig 6a to quantify mRNA levels of betaTrCP, FBXW2 or SKP2 where required.

Following reviewer's suggestion, we have performed these qPCR assays and quantified mRNA levels of all three genes. No significant difference was found, indicating the observed changes occurred at posttranslational levels due to altered ubiquitylation and degradation. The new results were included in these figures (Figs. 1f&g; 4g) with a new Supplementary Figure S6a generated to correspond to Figure 6a.

Figure 1h: The CHX assay does not seem to be working properly. This experiment needs to be repeated. Yes, we repeated this experiment and new data is presented (now Fig. 1j).

Figure 2d-2e: Endogenous FBXW2 should be analyzed.

Yes, per reviewer's suggestion, we have now replaced these figures by showing endogenous FBXW2 levels upon β TrCP overexpression in combination of GRK1 and VRK2 silencing (Figs. 2d&e).

Identification of the association between FBXW2 and SKP2 seems to be somewhat abrupt as it was identified through prediction based on the WB data of FBXW2 manipulated cells. Although the FBXW4 and FBXW8 are incorporated as a negative control in Supplementary Figure 4i-4j, it may be better to perform co-IP experiments using a panel of F-box proteins in order to demonstrate the specificity of the FBXW2-SKP2 interaction.

Per reviewer's suggestion, we have performed this experiment with newly generated results shown in Fig. S4a. Among a total of 8 F-box proteins tested, only SKP2 showed strong binding with FBXW2. A weak binding between SKP2 and β TrCP is likely mediated by FBXW2.

Figure 5a: The synchronization experiment should be performed using control- and betaTrCP-shRNA cell lines side by side to investigate how betaTrCP depletion affects the expression of FBXW2 and SKP2, through different cell cycle phases.

Per reviewer's suggestion, we have performed this experiment. Specifically, silencing β TrCP1 abrogates β TrCP1-dependent fluctuation of FBXW2. The SKP2 levels appear to be reduced substantially due to elevated levels of FBXW2. This newly generated data is now included in Supplementary Fig. 5a.

Figure 5: To validate an FBXW2 tumor suppressor function, a xenograft experiment should be carried out using a set of following cell lines: sh-Con, shFBXW2, and shFBXW2+shSkp2.

Per reviewer's suggestion, we have performed this experiment with newly generated results shown in Figure 5j&k. Specifically, as compared to shCon (group 1), shFBXW2 (group 2) significantly accelerated tumor growth/formation in nude mice, which, however, is completely abrogated by simultaneous silencing of SKP2 (group 3, shFBXW2+shSKP2). In fact, tumor growth rate was the lowest in group 3 due to SKP2 depletion (Fig. 5j&k) (page 14).

Figure 6a: The FBXW2 protein half-life should be analyzed and compared among multiple betaTrCP high- and low-expressing lung cancer cell lines to determine if the varying expression levels of FBXW2 is due to its altered half-life. Likewise, the half-life of SKP2 should be measured, but instead using multiple FBXW2 high- and low-expressing lung cancer cell lines.

Per reviewer's suggestion, we have performed this experiment with newly generated results shown in Supplementary Figure 6b. Specifically, we chose two lung cancer cell lines: A549 cells (β -TrCP-High/FBXW2-Low/SKP2-High) and H358 cells (β -TrCP-Low/FBXW2-High/SKP2-Low) and found that the half-life of these three proteins is largely correlated with their levels.

Responses to Reviewer #3 (Lung cancer and fbw proteins)

This manuscript outlines molecular interaction between three SCF-based F-box proteins in cancer. The authors indicate that a relatively new F-box protein, FBXW2, partakes within a Beta-TrCP-FBXW2-SKP2 axis that modulates lung cancer cell growth. The new contribution here really is the intermediary role of FBXW2 in cancer, as the roles of the other two F-box proteins in neoplasia (and lung cancer) are well characterized. The data presented incorporates biochemical interaction studies of the effectors, studies in lung tissue, and in vivo experiments. Some data as presented are not compelling for direct molecular interactions between F box proteins and several controls lacking.

1) The authors have not mapped the binding motifs between FBXW2 and SKP2.

Per reviewer's suggestion, we have performed this experiment with newly generated results shown in Supplementary Figure 4b&c. Specifically, we found that the C-terminal WD40 domain of FBXW2 (codons 139-454) binds to N-terminal domain of SKP2 (codons 1-150).

2) The authors suggest that BetaTrCP targets FBXW2 for polyubiquitination. The data are not compelling. First, there is no demonstration that endogenous FBXW2 is polyubiquitinated by the E3 complex. Second, the identification of an acceptor site is not demonstrated either within SKP2 or FBXW2. The type of polyubiquitination (K63/K48) is not identified.

To address reviewer's questions, we have performed following experiments as detailed below.

- 1) We used H1299 cells, which expressed a high level of FBXW2 and have now shown that endogenous FBXW2 can be polyubiquitylated by β TrCP1, but not β TrCP1 Δ F. This newly generated data is shown in Figure 2g.
- 2) The reviewer raised a great issue that we have not yet identified an acceptor site within either SKP2 or FBXW2. Given both proteins contain multiple lysine residues; it is not a trivial task to define their respective ubiquitylation site(s). Furthermore, it is also possible that artificial K->R mutation in one lysine residue may induce ubiquitylation to occur in adjacent site(s), making it difficult to define true ub site(s). Thus, we feel that this is a "nice-to-have" piece of data, but is not absolutely required for this study. We hope that the reviewer will agree with us. Thank you!
- 3) We have used K48/K63 or K48R/K63R ubiquitin mutants and fully defined that polyubiquitylation of both FBXW2 and SKP2 is via K48 linkage. These newly generated data are shown in Figure 2i and Figure 4n.

3) The SCF complexes recognize phosphodegrons and identity of a phospho-site within SKP2 that modulates interaction between FBXW2 as a binding partner and therefore polyubiquitination is not known.

Thank you for this valid point. Since we have now mapped the first 150 amino acids of SKP2 as the FBXW2 binding region, we compared the sequence of this region to a phosphor-peptide (codons 313-333) of GCM1, known to bind to FBXW2 (JBC 284:17411–17419, 2009), with special attention to the motif containing a string of serine/threonine residues. Indeed, we found an evolutionarily conserved motif of TSXXXS on codons 29-34 of SKP2 (Supplementary Fig.

4d). To test whether this is the motif mediating FBXW2-SKP2 binding, we mutated all three TSS residues to alanine (TSS→AAA) and found that mutant SKP2-3A is no longer bound to FBXW2 (Fig. 4b), nor ubiquitylated by FBXW2 (Fig. 4l) (pages 11-12). Thus, we postulated that the motif of TpSXXXpS found in both GCM1 and SKP2 is a putative consensus motif for FBXW2 binding.

Other concerns.

1) many Co-IPs lack controls, mention of % input etc. For example, Fig 1a another F-box subunit as a specificity control is lacking, a control sample where FBXW2 is depleted from cells before IP is not shown.

Per reviewer's suggestion, we have added controls with mention of % input throughout. For Figure 1a as well as Figure 4a, we include FBXL3 and FBXL11 as the negative control to show the specificity of β TrCP-FBXW2 binding and FBXW2-SKP2 binding. We also include a control sample where FBXW2 depletion leads to loss of its binding to both β -TrCP1 and SKP2. This piece of newly generated data is now shown here for reviewer's inspection, but not included in the manuscript due to space limitation.

2) Fig 1 h is not compelling. There is a 90% depletion of Beta-TrCP by knockdown and yet only a tiny increase in FBXW2.

This experiment was repeated and newly generated data is included in Fig. 1j.

3) Some panels in Fig 3 again would have been more convincing with inclusion of another F-box as a specificity control.

Per reviewer's suggestion, we added FBXW4 and FBXW8 as controls for FBXW2 in Figure 3g.

4) Figs 6b and 7h lack scale bars

We have now made scale bars more visible.

We feel that our manuscript has been significantly strengthened after these modifications, and thank you again for your thoughtful critiques.

REVIEWERS' COMMENTS:

Reviewer #1 (Remarks to the Author):

The authors have addressed all the concerns in my previous critique.

Reviewer #2 (Remarks to the Author):

The authors have addressed most of the concerns. The manuscript is suitable for publication.

Reviewer #3 (Remarks to the Author):

The revisions and responses by the authors generally address this reviewer's concerns. There is one minor clarification needed however. As I review the interaction studies in Supplemental Figure 4b,c between SKP2 and FBXW2, the discussion of the domains involved in mapping studies on the surface does not seem to make sense if I am reading the Figures and labeling correctly.

For example, "HA-FBXW2-N" presumably represents an HA-tagged F box fragment lacking the N-terminus (as suggested in the Figure legend). If this is so, then in Fig. 4c it would appear that ectopic expression of this plasmid followed by HA -pull down and immunoblotting with SKP2 antibody shows no signal binding, indicating that the N-term of FBXW2 is required for SKP2 interaction. Likewise, the HA-SKP2-C expressed plasmid results in lack of binding to FBXW2. Yet the authors state that " the C222 terminal WD40 domain of FBXW2 (codons 139-454) binds to N-terminal domain of SKP2 223 (codons 1-150) (Supplementary Fig. 4b&c).(page 11)". The authors should thus check their labeling or interpretation of data accordingly and also by extension discussion of results of phosphodegrons as needed. However if "HA-FBXW2-N" represents an HA-tagged F box fragment containing the N-terminus then I concur with their interpretation of data.

Dear Reviewer 3:

Below, we have addressed your remaining comments about Figure 4b&c.

The revisions and responses by the authors generally address this reviewer's concerns. There is one minor clarification needed however. As I review the interaction studies in Supplemental Figure 4b,c between SKP2 and FBXW2, the discussion of the domains involved in mapping studies on the surface does not seem to make sense if I am reading the Figures and labeling correctly.

For example, "HA-FBXW2-N" presumably represents an HA-tagged F box fragment lacking the N-terminus (as suggested in the Figure legend). If this is so, then in Fig. 4c it would appear that ectopic expression of this plasmid followed by HA -pull down and immunoblotting with SKP2 antibody shows no signal binding, indicating that the N-term of FBXW2 is required for SKP2 interaction. Likewise, the HA-SKP2-C expressed plasmid results in lack of binding to FBXW2. Yet the authors state that " the C222 terminal WD40 domain of FBXW2 (codons 139-454) binds to N-terminal domain of SKP2 223 (codons 1-150) (Supplementary Fig. 4b&c).(page 11)". The authors should thus check their labeling or interpretation of data accordingly and also by extension discussion of results of phosphodegrons as needed. However if "HA-FBXW2-N" represents an HA-tagged F box fragment containing the N-terminus then I concur with their interpretation of data.

First of all, we thank this reviewer for careful examination of our data. We apologize for unclear description on N-terminus-containing domain v.s. N-terminus-truncated domain. Indeed, HA- FBXW2-N represents an HA-tagged F box fragment containing the N-terminus, whereas HA- FBXW2-C represents an HA-tagged F box fragment containing the C-terminus. We have corrected this confusing description, and this portion of figure legend now reads

“H1299 cells were transfected with indicated plasmids expressing wild type or truncated proteins: FBXW2-F (full length, codons 1-454), FBXW2-N (N-terminus, codons 1-138), FBXW2-C (C-terminus, codons 139-454); SKP2-F (full length, codons 1-358), SKP2-N (N- terminus, codons 1-150), and SKP2-C (C-terminus, codon 151-358), followed by IP with HA-Ab and IB with antibodies against SKP2 (b) or FBXW2 (c). Expression of each plasmid was shown in the bottom panel”.

Thank you again for your thoughtful critiques which certainly made this manuscript much stronger.